# Transcranial focused ultrasound to V5 enhances human visual motion brain-computer interface by modulating feature-based attention

Joshua Kosnoff [1], Kai Yu [1], Chang Liu [1,2] & Bin He [1,3] ✉

A brain-computer interface (BCI) enables users to control devices with their minds. Despite advancements, non-invasive BCIs still exhibit high error rates, prompting investigation into the potential reduction through concurrent targeted neuromodulation. Transcranial focused ultrasound (tFUS) is an emerging non-invasive neuromodulation technology with high spatio-temporal precision. This study examines whether tFUS neuromodulation can improve BCI outcomes, and explores the underlying mechanism of action using high-density electroencephalography (EEG) source imaging (ESI). As a result, V5-targeted tFUS significantly reduced the error in a BCI speller task. Source analyses revealed a significantly increase in theta and alpha activities in the tFUS condition at both V5 and downstream in the dorsal visual processing pathway. Correlation analysis indicated that the connection within the dorsal processing pathway was preserved during tFUS stimulation, while the ventral connection was weakened. These findings suggest that V5-targeted tFUS enhances feature-based attention to visual motion.

The field of neuromodulation has witnessed the emergence of low-intensity transcranial-focused ultrasound (tFUS), a non-invasive brain stimulation technology with improved spatiotemporal specificity that promises to revolutionize our approach to understanding and modulating brain functions. While ultrasound modulation research dates back to the mid 1900s[1,2], a renaissance began with a 2013 chronic pain reduction pilot study in humans[3]. Since then, mechanistic studies and ultrasound parameter exploration have been conducted in various models, including in vitro rodent brain tissue[4–6], in vivo rodent models[7–11], non-human primates (NHP), and directly in humans.

NHP models have played a crucial role in advancing neuromodulation research, offering insights into complex cognitive functions. Notably, Deffieux et al. observed causal changes in antisaccade latencies through ultrasound stimulation of the frontal eye field (FEF)[12], Subsequently, Kubanek et al. demonstrated that FEF stimulation can bias choice behaviors[13]. Verhagen et al. found region-specific neural

effects of tFUS sonication at the supplementary motor area and frontal polar cortex, modulating connectional fingerprints of the targeted brain region with sustained effects[14]. Folloni et al. reported long-lasting, dissociable effects in the brain network by stimulating the amygdala and anterior cingulate cortex (ACC)[15]. Notably, behavioral and neural activities revealed that stimulating the deep brain area 47/12o and ACC impacted credit assignment choice and value representation in macaque subjects[16,17]. Zou et al. demonstrated the potential of the stimulation in treating brain functional disorders by inhibiting epileptiform discharge and reducing the frequency and duration of acute epileptic seizures in a penicillin-induced epilepsy NHP model[18]. Additionally, using functional magnetic resonance imaging (fMRI), Yang et al. observed bi-directional modulation of a specific somatosensory cortical area with tFUS stimulation[19]. As tFUS can be configured to precisely deliver stimulation to deeper brain areas non-invasively, Webb et al. observed that ultrasonic stimulation at the lateral

[1]Department of Biomedical Engineering, Carnegie Mellon University, Pittsburgh, PA 15237, USA. [2]Department of Biomedical Engineering, Boston, MA 02215, USA. [3]Neuroscience Institute, Carnegie Mellon University, Pittsburgh, PA 15237, USA. ✉e-mail: bhe1@andrew.cmu.edu

geniculate nucleus can persistently impact choice behavior, accompanied by increased gamma activity over the visual cortex, as demonstrated through intracranial electroencephalography (EEG)[20].

The tFUS renaissance has also spurred exciting developments in human neuroscience research[21], increasing its potential for addressing clinical challenges[21–23]. In addressing the medical challenge of disorders of consciousness (DOC), Cain et al. applied low-intensity focused ultrasound to the central thalamus in a cohort of DOC patients[24]. Despite reduced fMRI BOLD signals from the sonicated thalamus, the treatment resulted in significant recovery of neurobehavioral responsiveness in the patients. Yaakub et al. investigated the neurochemical basis of neuromodulation at the dorsal ACC and posterior cingulate cortex (PCC) in humans. They found that theta-burst ultrasound stimulation at the PCC can reduce inhibitory gamma-aminobutyric acid (GABA) levels, thereby altering excitability with sustained effects[25]. In a recent successful case study, Riis et al. demonstrated that reducing the activities of the subcallosal cingulate cortex with highly focused ultrasound neuromodulation targeting the deep brain region helps resolve treatment-resistant depressive symptoms[26].

This study investigates the potential of transcranial-focused ultrasound (tFUS) to enhance the human brain–computer interface (BCI). BCIs can read brain signals using either invasively implanted arrays or non-invasive techniques, such as scalp EEG, to perform tasks. Invasive BCIs have demonstrated the ability to control a robotic arm[27–30], decode speech[29,31,32], and interpret handwriting[33]. However, the need for brain surgery to implant the arrays limits their applicability. As a result, non-invasive BCIs offer a more feasible option for widespread application. Non-invasive BCIs have demonstrated the ability to control a robotic arm[34,35], operate a wheelchair[36], or type on a virtual keyboard[37,38]. Although non-invasive BCI research and development often involve healthy subjects, studies have shown that brain-decoding functionality remains intact in individuals with impairments[39], and can have positive impacts on those rehabilitating from strokes[40]. However, even with recent innovations, non-invasive BCIs still have online accuracy rates of approximately 70 to 80%[34,41–43], leaving room for improvement before widespread clinical adoption.

Traditional non-invasive neuromodulation devices, such as transcranial electric stimulation and transcranial magnetic stimulation, have limited spatial specificity, typically on the orders of centimeters[44], which may inadvertently modulate off-target brain areas[45]. The challenges of spatial and temporal resolutions typically restricted applications to occur before or in-between BCI sessions to facilitate cortical learning, rather than during the sessions[46–48]. Perhaps for these reasons, behavioral results for BCIs modulated by these devices are mixed[46–48]. tFUS, however, has demonstrated lateral precision on the order of millimeters[49]. Previous research has shown its capability to modulate cortical circuits[49–52] during stimulation sessions[49–51]. It has even been used in brain-brain interfaces, where information was read from one subject's brain via EEG, and corresponding tFUS-evoked perceptions were elicited in another subject[53]. However, to date, it has not been used to modulate the brain signals that directly control BCI. In the present study, we aimed to examine the effects of the neuromodulation on improving BCI control through human behavior and non-invasive electrophysiological recordings.

The BCI paradigm tested is a motion-onset visual evoked potential (mVEP) BCI speller, employing moving lines across a virtual keyboard to induce visual motion-based event-related potentials (ERPs)[37,42,54,55]. The primary ERP driving this paradigm is believed to be the N200, a negative deflection peaking 150 to 250 ms after stimulus onset, although the N100 (a negative deflection roughly 100 ms post-stimulus)[56] and the P300 (a positive deflection approximately 300 ms post-stimulus)[38] are also associated with visual processing and BCI spellers. A critical brain area involved in visual motion processing is V5, also known as the middle temporal complex[57–59]. A previous tFUS study

demonstrated that sonication of this area leads to improved visual motion detection[51].

In our study, we explore whether tFUS can be used to enhance BCI control by modulating V5 during an mVEP BCI speller task. We conduct neural data analysis at both the EEG sensor level and the brain region-specific source domain to examine possible neural mechanisms underlying the involvement in the performance of this BCI task.

## Results

### tFUS to the geometric center of V5 significantly reduces mVEP BCI speller errors

The Euclidean errors for each letter typed by each subject were calculated by assessing the spatial distance between the target letter and the typed letter. The experiment employed a cross-over design with four conditions: tFUS targeted at the geometric center of V5 (tFUS-GC), a non-modulated control representing a standard BCI without active tFUS (non-modulated), a decoupled-sham control with active but detached tFUS to account for potential auditory effects (decoupled-sham), and a spatial specificity control, where tFUS was steered to the geometric periphery of V5 near the inferior temporal gyrus (tFUS-GP). EEG source imaging of the non-modulated condition confirmed alignment with the reconstructed source at the center of V5 (Supplemental Fig. S1). The errors were analyzed using a linear mixed-effect model (Eq. 1), and a Type III analysis of variance (ANOVA) indicated significant differences in means across conditions ($p < 0.001$). Further condition-to-condition comparisons using $z$-tests and Bonferroni multiple comparisons $p$-value adjustment (Fig. 1a) revealed that the Euclidean error for the tFUS-GC condition ($N = 25$ subjects/356 trials; mean error $= 13.3 \pm 18.4\%$) was significantly lower than those of the non-modulated ($N = 25$ subjects/351 trials; mean error $= 15.5 \pm 18.7\%$; $p_{adjusted} < 0.01$), decoupled-sham ($N = 19$ subjects/268 trials; mean error $= 16.9 \pm 20.8\%$; $p_{adjusted} < 0.05$), and tFUS-GP conditions ($N = 16$ subjects/214 trials; mean error $= 17.0 \pm 18.2\%$; $p_{adjusted} < 0.001$). No significant difference was detected between the decoupled-sham, non-modulated condition, and tFUS-GP conditions ($p_{adjusted} > 0.05$). The effect size can be quantified with Cohen's $d$ value. The effect of tFUS-GC compared to all other controls is greater than 0.5 (Fig. 1b), indicating a moderate effect[60].

We further quantified the effect of experimental condition using Bayes Factor (BF) analysis (Fig. 1c). For the recommended default hyperparameter settings[61,62], the analysis found experimental condition to have a strong effect (median BF: 14.0; 95% confidence interval: 13.8 to 14.4) on Euclidean error. Rather than just present one BF corresponding to the recommended default hyperparameter, we present calculations of BF through a range of hyperparameter scaling up to "ultrawide" (1.0)[61]. These results provide a robust view that experimental condition has at least a moderate effect (BF > 3)[62], and often strong effect (BF > 10)[62] throughout nearly the entire range of hyperparameters scaling up to "ultrawide", thus providing a higher degree of confidence in the results than one singular value[63].

### tFUS to V5 significantly alters detected EEG sensor-level signals

For each experimental condition, preprocessed EEG decomposed into left posterior frequency bands and topographic maps filtered from 1 to 40 Hz are presented in Fig. 2a, b, respectively. A non-parametric permutation cluster repeated measures ANOVA test in MNE Python[64] was employed to compare data from subjects ($N = 13$) who completed all four experimental conditions. The results revealed a significant spatiotemporal cluster ($p < 0.05$) in the occipital regions between the four conditions over the span of 160 to 230 ms post stimulus onset (Fig. 2c).

### tFUS to the center of V5 significantly amplifies V5 N200 powers

To enhance the analysis, EEG data were further processed through electrophysiological source imaging (ESI)[65] using FreeSurfer[66,67] and MNE Python[64]. The data were $z$-scored with respect to the 200 ms pre-

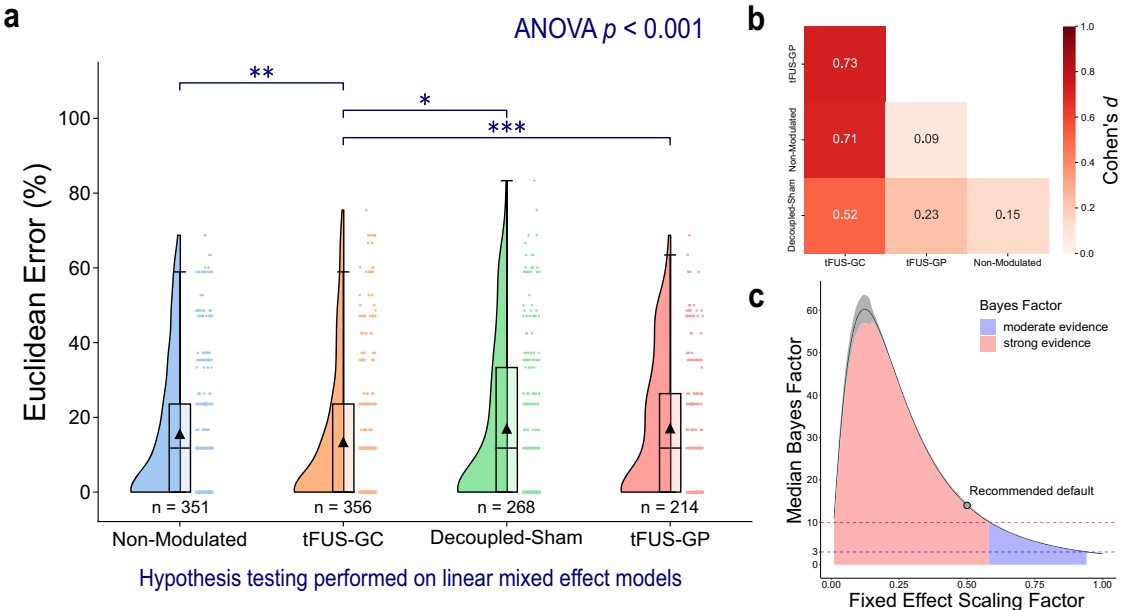

**Fig. 1 | tFUS to V5 significantly improves mVEP BCI speller outcomes.** Data were fit to a linear mixed-effect model to account for repeated measures across subjects and additional fixed effects from learning and fatigue (Eq. 1). **a** A raincloud plot of the BCI Euclidean error. The left portion of each subplot is a violin plot highlighting the distribution of the data. A box plot is at the center, marking the data median (center line), 1st and 3rd quartiles (whiskers = 1.5 * quartile ranges), and means (black triangle). The raw datapoints are presented to the right. Data are analyzed with the results of the linear mixed-effect models. A one-tailed ANOVA and a one-tailed *z*-test with Bonferroni correction were performed on the model's fit for each condition. The ANOVA test indicated significant differences ($p < 0.001$) in the mean Euclidean errors for each condition. tFUS sonication to the functional area located in the geometric center of V5 ("tFUS-GC"; *N* trials = 356 trials; mean error = 13.3 ± 18.4%; median error = 0.0%) led to a significantly lower Euclidean error for the mVEP BCI speller compared to non-modulated (*N* trials = 351; mean error = 15.5 ± 18.7%; median error = 11.8%), decoupled-sham (*N* trials = 268; mean error = 16.9 ±

20.8%; median error = 11.8%), and ultrasound steered to the geometric periphery of V5 ("tFUS-GP"; *N* trials = 214; mean error = 17.0 ± 18.2%; median error = 11.8%) conditions. No significant differences (significance level = 0.05) were found between the other three conditions. One-tailed *z*-test (with Bonferroni *p* adjustment) key: *$p_{adjusted} < 0.05$, **$p_{adjusted} < 0.01$, ***$p_{adjusted} < 0.001$. **b** The effect size was quantified using Cohen's *d* (Eq. 9). tFUS-GC had moderate ($d \geq 0.5$)[60] effects on BCI outcomes compared to all three control conditions. **c** The effect of experimental conditions was also quantified using Bayes Factors (BF). To provide a more robust view of the effect, the median ± 95% confidence interval (over 1000 iterations) BF was plotted over a range of fixed effect scaling factors, which provides a high degree of confidence in that experimental condition having a moderate (BF > 3)[62] to strong (BF > 10)[62] effect on the Euclidean error. For the recommended default value (scaling factor = 0.5), the BF was calculated to have a median of 14.0 and a 95% confidence interval from 13.8 to 14.4, which constitutes a strong effect. Source data are provided within the Source Data file.

---

stimulus baseline. N200 powers were computed by applying time-frequency Morlet waveform power transformation to the 100 to 250 ms post-stimulus window for theta (4–8 Hz), alpha (8–12 Hz), beta (12–30 Hz), and low gamma (30–40 Hz) frequencies (Fig. 3). Linear mixed-effect models were fitted to N200 powers as a function of experimental condition (Eq. 3). Comparisons between conditions revealed tFUS-GC (*z*-scores: theta power = 7.29 ± 5.97, alpha power = 5.59 ± 4.10, beta power = 3.82 ± 2.43, gamma power = 3.48 ± 2.20) to V5 during the mVEP BCI significantly amplified the N200 theta and alpha responses in the V5 region compared to the non-modulated (*z*-scores: theta power = 5.74 ± 4.36, alpha power = 4.75 ± 3.23, beta power = 3.51 ± 2.15, gamma power = 3.20 ± 1.92; $p_{adjusted} < 0.0001$ for theta band, $p_{adjusted} < 0.0001$ for alpha band), decoupled-sham (*z*-scores: theta power = 5.11 ± 3.74, alpha power = 4.38 ± 2.91, beta power = 3.32 ± 1.96, gamma power = 3.06 ± 1.79; $p_{adjusted} < 0.0001$ for theta band, $p_{adjusted} < 0.0001$ for alpha band), and tFUS-GP (*z*-scores: theta power = 6.83 ± 5.58, alpha power = 5.25 ± 3.79, beta power = 3.70 ± 2.38, gamma power = 3.35 ± 2.13; $p_{adjusted} < 0.05$ for theta band, $p_{adjusted} < 0.05$ for alpha band) conditions. Additionally, tFUS-GC beta and gamma N200 were significantly amplified over the non-modulated ($p_{adjusted} < 0.001$ for beta band, $p_{adjusted} < 0.001$ for gamma band) and decoupled-sham ($p_{adjusted} < 0.0001$ for beta band, $p_{adjusted} < 0.0001$ for gamma band) conditions, but not over tFUS-GP.

tFUS-GP was significantly amplified in comparison to the non-modulated ($p_{adjusted} < 0.0001$ for theta band, $p_{adjusted} < 0.001$ for alpha band, $p_{adjusted} < 0.05$ for beta band, $p_{adjusted} < 0.05$ for gamma band)

and decoupled-sham ($p_{adjusted} < 0.0001$ for theta band, $p_{adjusted} < 0.0001$ for alpha band, $p_{adjusted} < 0.001$ for beta band, $p_{adjusted} < 0.01$ for gamma band) conditions across all four frequency ranges. Additionally, the decoupled-sham condition was found to be significantly damped ($p_{adjusted} < 0.001$ for theta band, $p_{adjusted} < 0.001$ for alpha band, $p_{adjusted} < 0.05$ for beta band, $p_{adjusted} < 0.05$ for gamma band) compared to the non-modulated condition for all four frequency ranges.

## Theta and alpha N200 powers amplified by tFUS-GC are continued downstream in the dorsal pathway

The N200 theta power analysis was conducted for the superior parietal lobe (SP) and the inferior temporal gyrus (IT; Fig. 3), corresponding to the dorsal and ventral visual processing pathways, respectively[68–70]. Across all frequency bands, only the tFUS-GC condition (*z*-scores: theta power = 7.84 ± 6.27, alpha power = 5.79 ± 4.16, beta power = 3.79 ± 2.48, gamma power = 3.38 ± 2.19) exhibited significant differences in the SP compared to other conditions. Theta and alpha N200 powers in tFUS-GC were significantly amplified compared to the non-modulated (*z*-scores: theta power = 6.78 ± 5.19, alpha power = 5.30 ± 3.63, beta power = 3.54 ± 2.19, gamma power = 3.16 ± 1.93; $p_{adjusted} < 0.0001$ for theta band; $p_{adjusted} < 0.01$ for alpha band), decoupled-sham (*z*-scores: theta power = 6.62 ± 4.98, alpha power = 5.34 ± 3.66, beta power = 3.71 ± 2.27, gamma power = 3.31 ± 2.00; $p_{adjusted} < 0.0001$ for theta band; $p_{adjusted} < 0.01$ for alpha band), and tFUS-GP (*z*-scores: theta power = 6.96 ± 5.30, alpha power = 5.29 ± 3.60, beta power = 3.66 ±

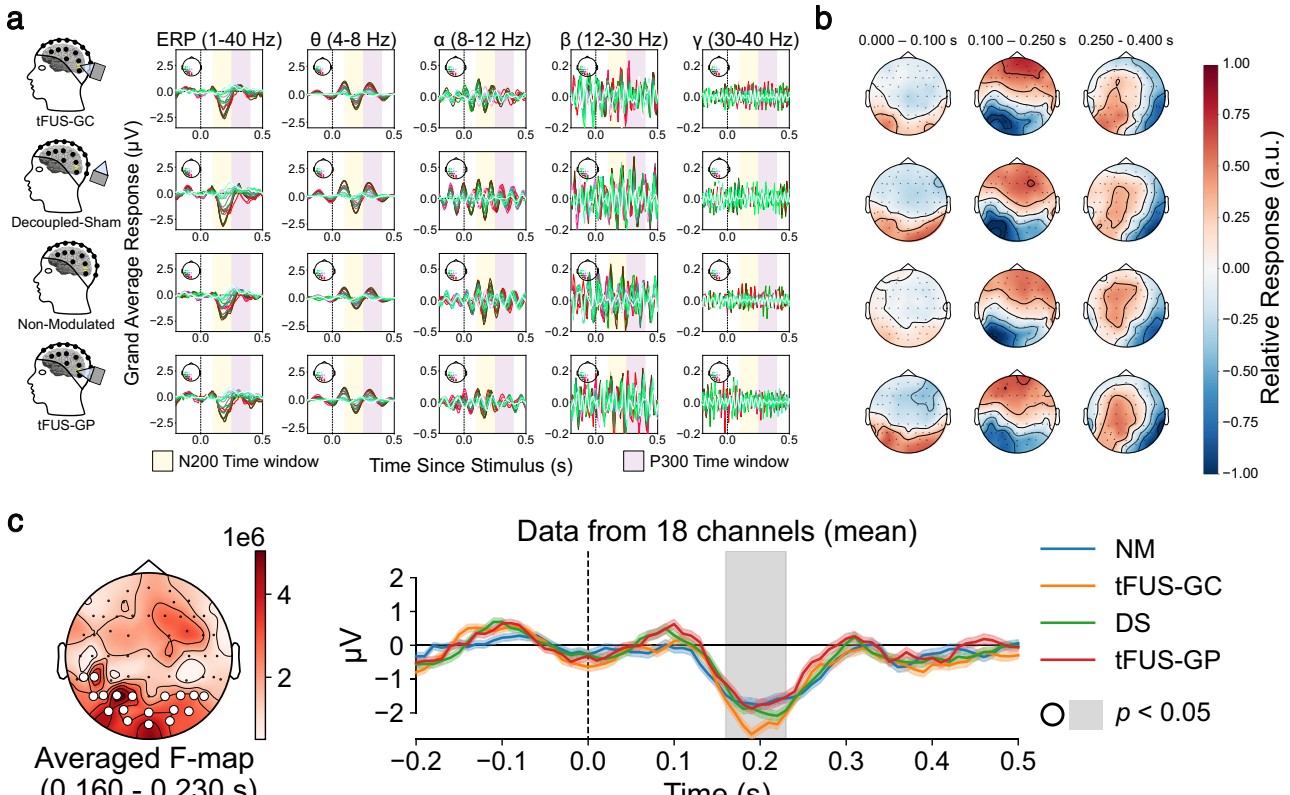

**Fig. 2 | Significant differences are found between conditions in the EEG sensor domain. a** Averaged left posterior electrode responses for 1 to 40 Hz (graph column 1), theta frequencies (graph column 2), alpha frequencies (graph column 3), beta frequencies (graph column 4), and gamma frequencies (graph column 5) for tFUS-GC (top), decoupled-sham (second row), non-modulated condition (third row), and tFUS-GP (bottom). Yellow and purple highlight the approximate N200 and P300 waveform responses within the 100 to 250 ms and 250 to 400 ms windows, respectively. **b** Topographic maps of the trial's averaged activity for 0 to 100 ms (left), 100 to 250 ms (middle) and 250 to 400 ms (right) post stimulus filtered between 1 to 40 Hz for tFUS-GC (top), decoupled-sham ("DS"; second row), non-modulated ("NM"; third row), and tFUS-GP (bottom) conditions. The topo colormap is scaled to the relative max/min response for each. **c** A significant spatiotemporal cluster ($p < 0.05$) between the 1 to 40 Hz filtered conditions using MNE Python's non-parametric spatiotemporal cluster test with 1000 permutations (test: one-tailed repeated measures ANOVA) when comparing all four conditions. (left) The F-statistics of a significant spatial cluster denoted by white circles over the electrodes. (right) The mean activity across channels and trials of the four trial conditions ($\pm 1$ standard deviation). Gray shaded regions indicate a significant ($p < 0.05$) temporal cluster corresponding to the spatial cluster.

2.28, gamma power = $3.27 \pm 2.01$; $p_{adjusted} < 0.0001$ for theta band; $p_{adjusted} < 0.01$ for alpha band) conditions. Beta and gamma N200 powers in tFUS-GC were only significantly amplified compared to the non-modulated ($p_{adjusted} < 0.05$ for beta band; $p_{adjusted} < 0.05$ for gamma band).

When examining the ventral pathway (IT), no significant difference was found among the experimental conditions of tFUS-GC (z-scores: theta power = $4.91 \pm 3.60$, alpha power = $4.01 \pm 2.60$, beta power = $3.16 \pm 1.93$, gamma power = $2.92 \pm 1.77$), non-modulated (z-scores: theta power = $4.79 \pm 3.67$, alpha power = $4.02 \pm 2.72$, beta power = $3.15 \pm 1.92$, gamma power = $2.93 \pm 1.78$), and decoupled-sham (z-scores: theta power = $5.04 \pm 3.71$, alpha power = $4.29 \pm 2.88$, beta power = $3.23 \pm 1.92$, gamma power = $2.99 \pm 1.77$) across any of the four frequency bands. In the tFUS-GP condition, however, powers in the theta, beta, and gamma frequencies were significantly downmodulated (z-scores: theta power = $4.41 \pm 3.26$, alpha power = $3.75 \pm 2.57$, beta power = $2.95 \pm 1.89$, gamma power = $2.75 \pm 1.76$) compared to those in the decoupled-sham ($p_{adjusted} < 0.01$ for theta band, $p_{adjusted} < 0.05$ for beta band, $p_{adjusted} < 0.05$ for gamma band), non-modulated ($p_{adjusted} < 0.05$ for theta band, $p_{adjusted} < 0.05$ for beta band, $p_{adjusted} < 0.05$ for gamma band), and tFUS-GC ($p_{adjusted} < 0.05$ for theta band; $p_{adjusted} < 0.05$ for beta band; $p_{adjusted} < 0.05$ for gamma band) conditions. In addition, the alpha power in tFUS-GP was

significantly downmodulated compared to the decoupled-sham ($p_{adjusted} < 0.01$) and non-modulated ($p_{adjusted} < 0.05$).

## tFUS-GC maintains dorsal connection while weakening ventral pathway

Cortical connections were quantified using the absolute Pearson's correlation coefficient (r) for all four conditions in both theta and alpha frequency ranges. The correlation coefficients for each condition in each frequency range were fit using a linear mixed-effect model and examined with a one-tailed z-test with false-discovery rate multiple comparison correction. Results indicate that the correlation between V5-IT is significantly lowered ($p_{adjusted} < 0.05$) in both the tFUS-GC and the decoupled-sham conditions compared to the non-modulated condition in the theta band (Fig. 4g), and compared to both the non-modulated and the tFUS-GP condition in the alpha band (Fig. 4i). In the alpha frequency range, there was no significant difference across conditions in the dorsal pathway connection ($p_{adjusted} > 0.05$, Fig. 4h). This implies that the dorsal pathway is not significantly changed by ultrasound neuromodulation in the alpha band. In the theta range, however, the decoupled-sham condition significantly weakened the dorsal pathway compared to all three other conditions (Fig. 4f), indicating that the potential audible sound presented in the decoupled-sham condition may reduce visual motion attention, thus weakening

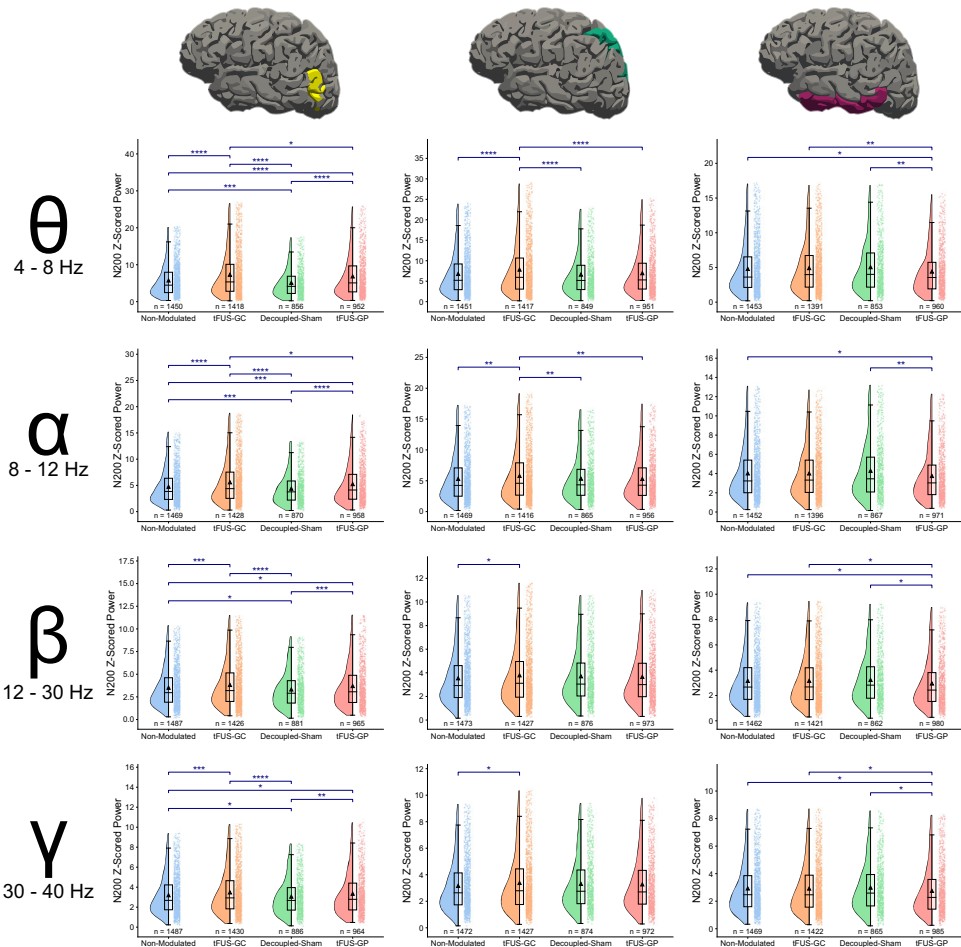

Hypothesis testing performed on linear mixed effect models

**Fig. 3 | tFUS-GC N200 modulation is conveyed through the dorsal pathway in theta and alpha frequencies.** Raincloud plots of the N200 power are presented at various frequencies and locations. The left portion of each subplot is a violin plot highlighting the distribution of the data. A box plot is at the center, marking the data median (center line), 1st and 3rd quartiles (whiskers = 1.5 * quartile ranges), and means (black triangle). The raw datapoints are presented to the right. Data are analyzed with the results of the linear mixed-effect models. EEG source imaging of V5 reveals a significant amplification of the N200 theta and alpha power for tFUS-GC modulation compared to decoupled-sham, non-modulated, and tFUS-GP control conditions. There is also significant damping of the decoupled-sham compared to the non-modulated condition. In beta and gamma frequencies, tFUS-GC exhibits amplified power compared to the non-modulated and decoupled-sham conditions, but not tFUS-GP. EEG source imaging of the superior parietal lobe shows significant N200 theta and alpha amplification in the tFUS-GC condition compared with the non-modulated, decoupled-sham, and tFUS-GP conditions. In beta and gamma frequency ranges, this amplification is only observed compared to the non-modulated. tFUS-GC is not significantly amplified compared to any of the control conditions in IT across any of the frequency ranges. One-tailed $z$-test key false-discovery rate adjustment: $*p_{adjusted} < 0.05$, $**p_{adjusted} < 0.01$, $***p_{adjusted} < 0.001$, $****$ $p_{adjusted} < 0.0001$. Source data are provided within the Source Data file.

both the dorsal and ventral connections of the visual processing in the brain.

## A natural language processing-based shared control algorithm further improves the BCI speller usability

Initial testing of a Shared Control autoRegressive Integrated Bayesian Estimator (SCRIBE; Supplemental Fig. S2, Supplemental Movie S1, Supplemental Movie S2) suggests a shared control word-correction program can be used to further improve the software component of BCI spellers, especially for naïve human subjects, even when the typed accuracy is as low as 50%. While Bayesian autocorrection using word frequency is not a new concept for BCI spellers[71], our algorithm considers both word frequency as well as keyboard-specific Euclidean error of letters. Further, existing Bayesian autocorrection formulas have considered just the current word[71], while the presented SCRIBE considers words in the context of a sequence, thus imposing additional linguistic meaning onto the output. The current iteration only considers up to two-word sequences (autoregressive order 1), but there is

available data on Google N Gram to consider up to five-word sequences (autoregressive order 4).

## Ex-vivo measurements of the applied tFUS parameters

tFUS target pressure and derated intensities generated by the customized 128-element random array ultrasound transducer H275 were estimated by transcranial pressure readings through 3D needle hydrophone scanning behind a fully hydrated real human skull sample (ethnicity: Caucasian, age: 58, gender: male) submerged in degassed water in a water tank. The following ultrasound parameters are measured and estimated for the administered transcranial ultrasound stimulation: peak-to-peak pressure at the brain target 0.2 MPa, derated spatial-peak temporal-average intensity ($I_{SPTA.3}$) 101 mW/cm², and derated spatial-peak pulse-average intensity ($I_{SPPA.3}$) 0.168 W/cm². While currently there are no FDA safety guidelines specifically for ultrasound neuromodulation, the estimated ultrasound energy is within safety limits for ultrasound diagnostic imaging ($I_{SPTA.3} < 720$ mW/cm², $I_{SPPA.3} < 190$ W/cm²)[72].

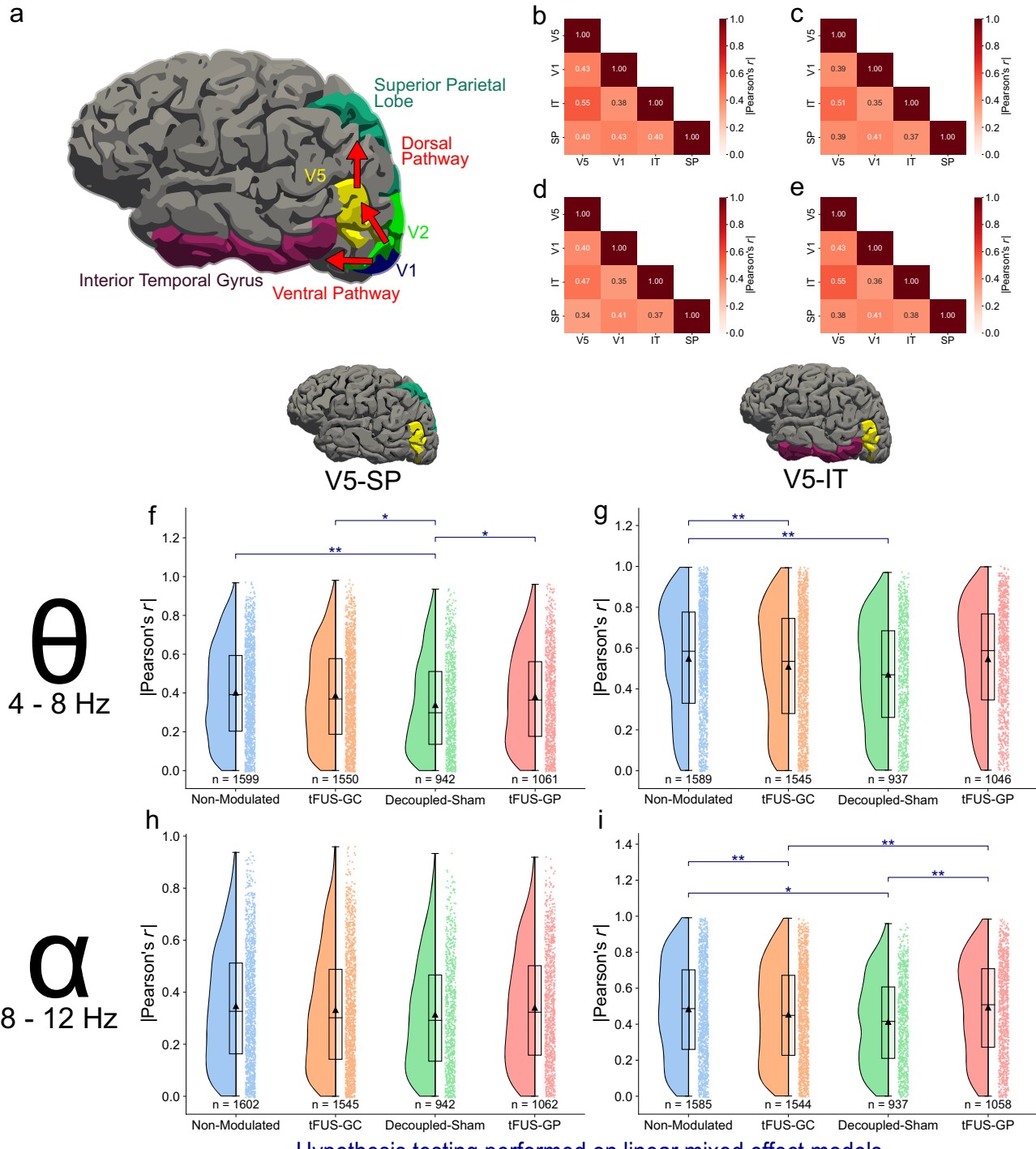

**Fig. 4 | tFUS-GC's effects are conveyed through the theta frequency of the dorsal pathway. a** The canonical dorsal and ventral visual processing pathways with their FreeSurfer associated labels. **b**–**e** Grand-average |Pearson's r| values for theta frequency absolute correlation for **b** non-modulated, **c** tFUS-GC, **d** decoupled-sham, and **e** tFUS-GP of the mVEP BCI epochs. **f**–**i** Raincloud plots of the correlation coefficient in theta-alpha frequencies for dorsal and ventral pathways. The left portion of each subplot is a violin plot highlighting the distribution of the data. A box plot is at the center, marking the data median (center line), 1st and 3rd quartiles (whiskers = 1.5 * quartile ranges) and means (black triangle). The raw datapoints are presented to the right. Data are analyzed with the results of the

linear mixed-effect models. **f** The dorsal pathway (V5-SP) theta frequency band correlation is significantly reduced for the decoupled-sham condition compared to the others. **g** The ventral pathway (V5-IT) theta frequency band is significantly decorrelated for both the tFUS-GC and decoupled-sham compared to the non-modulated condition. **h** There is no significant difference across alpha frequency correlations for any of the conditions in the dorsal pathway. **i** tFUS-GC and the decoupled-sham alpha frequency bands are significantly decorrelated compared to the non-modulated and tFUS-GP conditions, but not from each other. One-tailed z-test with false-discovery rate correction key: $^*p_{adjusted} < 0.05$, $^{**}p_{adjusted} < 0.01$. Source data are provided within the Source Data file.

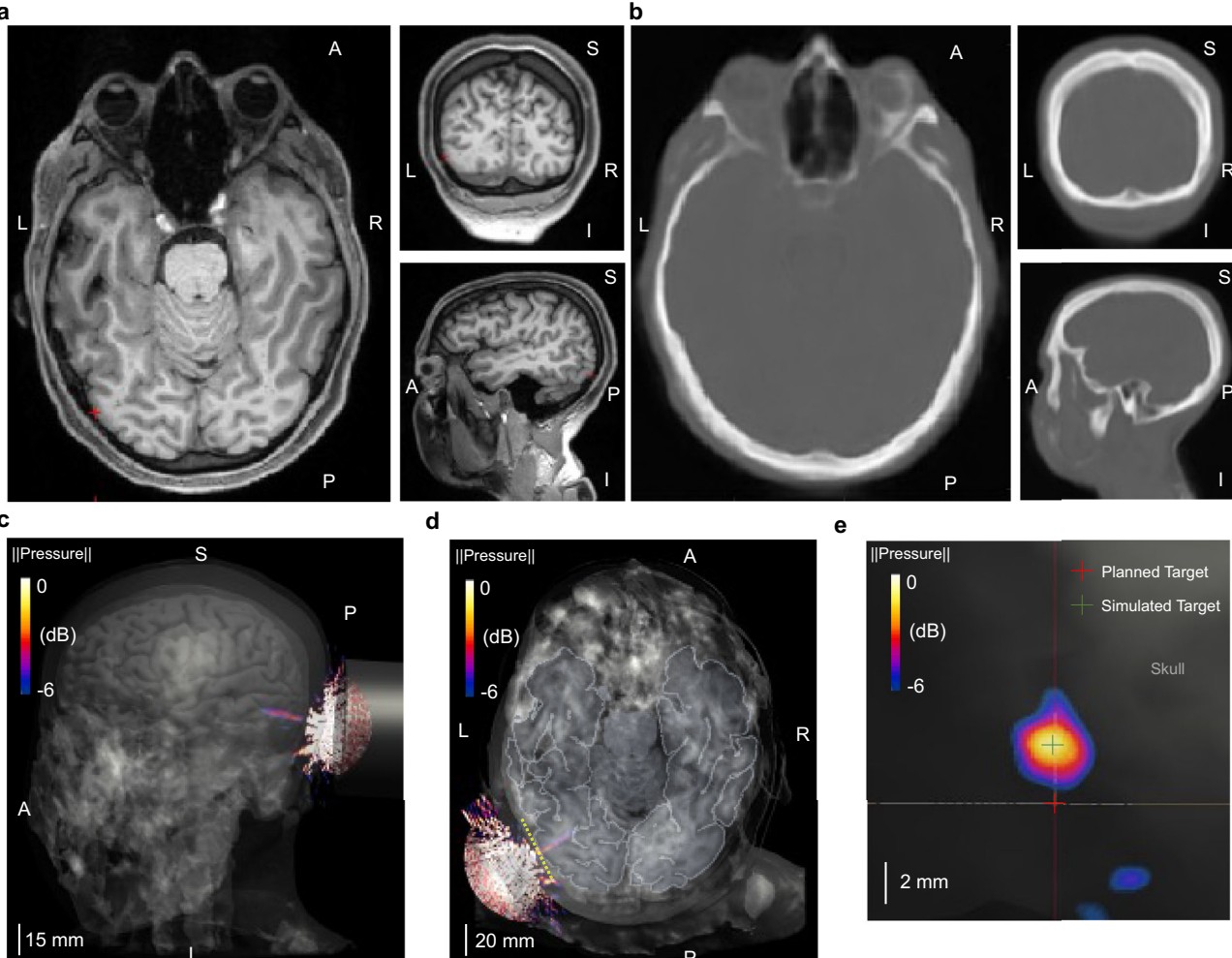

**Fig. 5 | Computer simulations of tFUS generated by the 128-element random array ultrasound transducer in a representative individual (Subject #5). a** The subject-specific T1-weighted structural magnetic resonance images in horizontal, coronal and sagittal planes, with the red cross indicating the geometrical center of the V5 area. **b** Pseudo-CT images generated from the MRI in the same imaging planes as illustrated in **a**. **c**, **d** The transcranial pressure field and focused ultrasound beam (−6dB focal pressure volume) co-registered with the subject skull and brain models. **e** A lateral view of the transcranial ultrasound focus (−6dB focal pressure volume) on the surface of V5 along the yellow dashed line in **d**. The center of planned target used for optical-based brain navigation is depicted using a red cross, while the center of simulated ultrasound target is at the location marked with a green cross. The deflection of ultrasound focal spot is quantified with the spatial distance between these two crosses, which is measured as 2.84 mm for this subject. Extensive subject-wise characterizations of transcranial ultrasound focus are depicted and listed in Supplemental Fig. S3 and Supplemental Table S1.

## In-silico subject-specific tFUS simulations show minimal ultrasound distortions

Although the ex-vivo scanning provided direct characterizations of the transcranial-focused ultrasound beam, it may not reflect the specific skull features of our participants, potentially overlooking possible distortions due to varied skull properties among all the human subjects in the study. For this reason, we derived pseudo-CT images for subject-specific skull models from their T1-weighted MRI (Fig. 5a). After segmenting the generated pseudo-CT images (Fig. 5b) and assigning specific mechanical properties to the segmented brain tissues and skull bones, realistic tFUS computer simulations were implemented to estimate possible ultrasound distortions behind individual skull models. In the simulations, we placed the computational model of H275 over the subject's head to target at the geometrical center of V5, as in our human experiments.

As illustrated in Fig. 5c, d, the focused ultrasound beam was located at the targeted V5 area in a representative participant without significant aberrations. While the ultrasound focal spot at the surface of the V5 is not significantly distorted, we observed that there was deflection (approximately 2.84 mm) from the center of the planned stimulation site (Fig. 5e). Even with that deflection, the tFUS-GC condition still delivered ultrasound stimulation to the functional area of V5 (the functional area for this paradigm was found to be approximately 15 mm in radius; Supplemental Fig. S1). We provided all the focus deflection measurements in Supplemental Fig. S3a, b and included visualizations of the six worst cases of tFUS beam deflection in Supplemental Fig. S3c. Overall, the subject-specific tFUS simulations show that the transcranial ultrasound beams were localized at the V5 area without significant distortions or standing wave pattern. In Supplemental Table S1, we provided further subject-wise characterization of the axial shift of transcranial ultrasound focus from the skull-brain interface, and the estimations of the pressure ratios (dividing the maximum pressure within the brain by the maximum pressure in the head), derated spatial-peak temporal-average intensities, and derated spatial-peak pulse-average intensities in the brain. The average axial shift is 4.65 mm, and the mean value of the pressure ratios is 0.14. All subject-specific derated intensities were concordant with the aforementioned FDA safety guidelines.

## Discussion

Our study demonstrates that low-intensity transcranial-focused ultrasound (tFUS) neuromodulation can enhance the performance of a visual motion-based brain–computer interface through significant theta and alpha power amplifications of the dorsal visual processing pathway. tFUS at 0.2 MPa peak-peak pressure and 3 kHz pulse-repetition frequency was administered to a subject's left hemisphere V5 while they typed letters through a virtual keyboard presented on a screen with an mVEP BCI speller. Euclidean errors of this BCI speller in the tFUS-GC (tFUS directed to the functional area located at the geometrical center of V5, brain source estimated using EEG source imaging and presented in Supplemental Fig. S1) condition were significantly lower than those in non-modulated, decoupled-sham, and tFUS-GP (tFUS applied to the geometrical periphery of V5 near IT; Supplemental Fig. S4) conditions. In addition to the statistical significance, the quantified effect size of Cohen's $d$ indicates tFUS-GC had a moderate effect compared to the other conditions (Fig. 1b). Though Bayes Factor analysis does not provide specific insight into intra-condition comparisons, it does robustly corroborate that the experimental condition has a moderate to strong effect on BCI Euclidean error (Fig. 1c). Source analyses of the ultrasound-targeted left hemisphere V5 parcellation (tFUS-GC) exhibit significantly amplified N200 theta and alpha powers compared to the other three conditions. This amplification is observed downstream through the dorsal visual processing pathway, as the significantly amplified N200 theta and alpha powers in the tFUS-GC condition are also apparent in the superior parietal lobe. Correlation analysis supports that the V5-superior parietal lobe information flow is maintained from the non-modulated to the tFUS-GC condition, while the correlation with the ventral pathway is significantly reduced. These results, contextualized with the previous alpha[73–75] and theta[76–81]-band research, indicate that V5-targeted tFUS neuromodulation may increase visual motion feature-based awareness.

The EEG sensor domain-level analysis employed a non-parametric permutation cluster test to address the multiple comparisons problem. It is important to note that significant clusters, as identified by the test ($p < 0.05$), indicate overall differences among the conditions, but do not guarantee significant differences for every time point within the cluster; rather, they suggest an average difference across the cluster[82]. There were differences in EEG sensor locations among subjects, as EEG caps were not centrally aligned with subject landmarks, but individually rotated for optimal exposure of the subject's V5 to the ultrasound transducer. To minimize the impact of these differences and ensure reliable EEG source analyses, we digitized the positions of EEG electrodes for each human subject. The captured EEG electrode placements were then used for subject-specific head models and source computations across various brain parcellations. Minimum norm estimation[83] was employed to solve the EEG source imaging problem, and high-density EEG caps with 62 active recording electrodes provided sufficient spatial sampling[84]. To simplify the multiple comparison dimensionality, we aggregated theta, alpha, beta, and low gamma power over the N200 time window from multiple electrodes, allowing for direct comparisons across conditions with linear mixed-effect models.

The EEG source analyses focused on the theta, alpha, beta, and low gamma components of the N200. Examining this window for the V5 area revealed significantly amplified N200 powers in both theta and alpha frequency ranges in the tFUS-GC condition compared to the three control conditions. In the beta and low gamma frequency ranges, tFUS-GC N200 power was significantly up-modulated relative to the decoupled-sham and non-modulated conditions, but not in comparison to tFUS-GP. These findings indicate that, while the presence of ultrasound modulates all four frequency ranges, the theta and alpha responses specifically convey information about whether the ultrasound was targeted at the center of V5 or the periphery.

Extending this analysis downstream the dorsal processing pathway to the SP area corroborates these findings, consistent with the known dorsal pathway connections from V5 to the parietal lobe[68,85,86]. N200 powers in theta and alpha frequencies for tFUS-GC are significantly amplified compared to the three control conditions, while in the beta and gamma ranges, significant amplification is observed only relative to the non-modulated condition. Given the BCI speller's behavioral results showing significant improvement with tFUS-GC over all three controls, and a similar trend in the underlying neural activity, it confirms the overall higher relevance of alpha and theta activities to this BCI paradigm compared to higher frequency bands. Meanwhile, the N200 theta power analysis of the inferior temporal lobe reveals no significant difference among conditions, aligning with the fact that visual motion information from V5 typically does not project to the ventral pathway[68,87].

The specific mechanisms underlying focused ultrasound neuromodulation are still being investigated[4,88–90]. One early hypothesis suggested that tFUS increases local interneuron firing, influencing the flow of information[49]. This aligns with our study's observation of increased alpha power, as alpha oscillations are often associated with inhibitory, top-down attention processes[73–75]. More specifically, Payne et al. found that increased alpha activity was linked to an improved ability to ignore distracting information[74]. Combining these findings supports the idea that tFUS enhances target-location-specific activities by filtering out distracting information.

tFUS in our BCI paradigm not only significantly amplifies alpha power but also theta power (Fig. 3). The role of theta power in visual attention seems to be somewhat controversial, as it has been reported that occipital theta is both decreased[91] and increased[76] with attention to visual stimuli. We address this inconsistency by first focusing on the enhanced tFUS-GC theta power in the superior parietal lobe. There is a larger consensus that increased theta power in the parietal region is tied to increased attention[76–79]. Therefore, tFUS-GC may increase feature-based attention downstream the dorsal pathway, at the very least. It would be counterintuitive for tFUS to decrease attention-related activities at its site of sonication while only to increase it later downstream, so we view it as far more likely that the observed heightened occipital theta power is also linked to increased attention.

Pairing the effects on both theta and alpha frequencies together, our results suggest that the mechanism of action of tFUS in our BCI paradigm is two-fold: (1) it increases local attention by increasing the theta power and (2) it filters out external information by boosting alpha power.

From the N200 power analysis (Fig. 3), only theta and alpha bands exhibited significant differences between the tFUS-GC condition and the other three control conditions at the superior parietal lobe. Since the behavior results also indicated a significant difference between tFUS-GC and all three controls, it is expected that there are underlying different neural effects between tFUS-GC and all other three controls as well. For this reason, we focus connectivity analyses on the theta and alpha bands. A correlation analysis of the ventral pathway theta bands (Fig. 4g) reveals a significant disconnection ($p < 0.01$) in both the tFUS-GC and the decoupled-sham condition compared to the non-modulated condition. Since the only difference between the decoupled-sham and the non-modulated condition is the audible sound, it follows that this attenuated connection is likely due to potential distraction produced by ultrasound pulse-repetition frequency (PRF)-related sound[92]. However, there is no significant difference between tFUS to the geometric periphery of V5/IT (tFUS-GP) and the non-modulated, indicating that tFUS effects at the location along the border of V5-IT may compensate the sound-distraction-induced disconnection. This pattern is also observed in the alpha frequency range, where tFUS-GC and the decoupled-sham are significantly disconnected compared to the non-modulated and tFUS-GP, while tFUS-GP and the non-modulated conditions do not show a significant difference (Fig. 4i).

In the dorsal pathway (Fig. 4f), the decoupled-sham condition induced significantly less connection in the theta band than in the other three conditions. This suggests that the distraction caused by audible sound is responsible for the decreased correlation. There are no significant differences either between tFUS to the geometric center of V5 (tFUS-GC) and the non-modulated condition or between tFUS to the V5/IT periphery and the non-modulated condition. This indicates that both stimulation effects delivered to the corresponding brain locations may potentially compensate for the sound-distraction-induced disconnection and reconnect the pathway. There is no significant difference across conditions in the alpha band (Fig. 4h), indicating that potential auditory effects do not affect dorsal connectivity within the alpha frequency range in this paradigm. The lack of significant differences also suggests that tFUS itself does not modulate dorsal pathway connectivity in the alpha frequency range.

In addition to the aforementioned direct effects of alpha[73–75] and theta[76–79] frequency bands on attention, there is a growing understanding that the brain's theta rhythm itself may hold information about attention[80,81]. Studies have shown that the brain spatially samples attention at the theta frequency rate[80,81]. Therefore, breaking the theta band correlation from one region to another may be a direct method of disrupting attention sampling. This suggests that tFUS not only amplifies attention through alpha and theta power modulation in the dorsal pathway but also that the decorrelating of the theta-band activity in the ventral pathway may impair the sampling ability from this pathway.

More evidence of auditory effects on the brain can be seen in the source analysis of V5 N200 power (Fig. 3), where the decoupled-sham N200 power is significantly dampened compared to that of the non-modulated condition in all four frequency bands. The results of both connectivity and power analyses might stem from the ERP response being influenced by attention[93]. The beeping noise generated by the PRF of the decoupled-sham ultrasound may have distracted the subjects, without the actual ultrasound modulation effects delivered in the tFUS condition to counteract the distraction. It is well-accepted in the tFUS community that noise alone can induce neurological responses[94]. However, researchers hold different views on how significant a role sound-based modulation plays. Some assert that the noise has minimal effect on the modulation[15], and mitigating it does not affect the outcome of motor responses[95]. Others contend that the entirety of tFUS-based neuromodulation is due to auditory side effects[96,97]. The results from our study acknowledge the target-specific tFUS neuromodulation effects and the impact of auditory-related brain responses. Furthermore, our correlation results indicate that, at least in the case of attention-based tasks, the auditory effect may cause widespread circuitry disconnection, with tFUS selectively reconnecting pathways based on the stimulation targets. Future research is warranted to better understand ultrasound auditory confounds in BCI paradigms.

tFUS neuromodulation was implemented concurrently during the BCI task, meaning it modulated brain signals while the visual stimuli were presented. This differs from traditional non-invasive neuromodulation studies for BCI, which are typically applied before or after the trials to either prepare the brain or consolidate learning[46–48]. The technique in this study represents a step towards more closely mimicking invasive electrical stimulation[98]. As noted in the methods section, we applied 1–4 repeats of sonications to each subject per BCI epoch. We analyzed the differences in corresponding subject-average tFUS-GC Euclidean error as a function of number of sonications and found no significant difference (Kruskal-Wallis; $p = 0.425$). This may be due to the skewed sample sizes (18 of the 25 subjects received four sonications per epoch), and future studies are warranted to investigate the exact effect and optimization of sonication numbers.

In power and correlation analyses, we reconstructed brain activities at V5 using EEG source imaging. This involved utilizing signals recorded from all other 62 recording electrodes to reserve physical space for positioning the ultrasound transducer over V5. EEG source imaging estimates brain electrical activity from a multitude of electrode recordings over the scalp, relying on the quantitative relationship between brain sources and the scalp field manifestation[65]. Brain activity at a given region is thus reflected by multiple electrodes over the scalp, not just the most adjacent electrode. What is crucial is sufficient spatial sampling[84], which allowed us to accurately estimate brain activity from the target brain region.

One limitation of this study is related to the calculation of N200 powers. The process involved computing the mean power within each frequency band in the 100 to 250 ms post-stimulus time window. Since N200 peak times are commonly found between 150 and 250 ms post onset[99–102], an earlier time window was used to capture the onset of the wave. However, this may lead to potential overlap with the N100 ERP[56] and the end of the window may overlap with the P300 ERP[103]. Visualization of the trial averages (Fig. 2a) indicates that, on average, this window corresponds to an N200 waveform. However, this may not be guaranteed for every trial. Future works may explore automated ERP differentiation and characterization to best isolate one from another. Despite this limitation, given the large number of trials, exclusion of statistical outliers, and the equal likelihood of this issue across all trial conditions, the results still provide an accurate representation of N200 power. Additionally, our focus was on within-frequency analysis in individual EEG bands power, specifically theta, alpha, beta, and gamma, rather than exploring cross-frequency interactions. Future investigations can examine how tFUS modulates cross-frequency interactions in EEG.

Another limitation of our study is that we focused solely on the accuracy of the BCI speller output. Information transfer rate (ITR) is another popular metric used to quantify BCI speller performance, which relates to the speed of operation[42,54,55]. In our paradigm, we used a constant speed for each subject by using the same number of scan repeats across conditions without any early stopping for highly probable classifier predictions. Due to the current BCI task design, we are unable to assess how tFUS neuromodulation may affect the speed of the spelling task. Future work may further investigate how tFUS neuromodulation of BCI spellers changes the ITR.

To address the multiple comparison problems, we used Bonferroni and false-discovery rate (FDR) multiple comparison corrections. We used Bonferroni correction, which is more conservative, to rigorously investigate the BCI behavior error outcomes, as the effect of the on BCI performance was the primary focus of this investigation and we wanted to be extra conservative in avoiding potential false positives (type I errors). For the neural data analysis, we used FDR, which is a less conservative method that can better balance the risk of possible type I and type II errors.

Our subject cohort consisted of young and healthy individuals. Since an important goal of BCI is to aid in function restoration or rehabilitation in paralyzed or otherwise impaired subjects, future work should investigate the application of tFUS-modulated non-invasive BCI on additional cohorts of subjects. This may include studies exploring direct clinical translations on impaired subjects and/or older populations.

While administering low-intensity tFUS stimulation is generally considered safe, we monitored the safety of tFUS for our human subjects during the experimental sessions. To assess potential adverse effects, we administered a concise questionnaire to participants before and after each tFUS/control session. This questionnaire was designed to capture reports of symptoms such as headache, neck pain, dental pain, nausea, dizziness, anxiety, and abnormal muscle contractions, among others. It is noteworthy that none of our human subjects reported any adverse symptoms after the the sessions.

In conclusion, our results demonstrate that the neuromodulation to human V5 significantly reduces errors in mVEP-based BCI spelling. This is quantified with traditional $p$-value analysis, Cohen's

*d* effect size, and Bayesian Factor analysis, all of which concur that the effect of the neuromodulation on the V5 is medium to strong. We also investigate the underlying neurological changes induced by tFUS and sham conditions to provide a thorough framework for the underlying mechanism. Concurrent neural electrophysiological recordings indicate that in the mVEP-based BCI, tFUS neuromodulation causes a significant amplification of theta and alpha powers in the immediate V5-targeted area, as well as downstream in the dorsal vision processing pathway. Connectivity analysis, along with previous research on alpha-theta power, provides critical evidence that the increased theta power due to concurrent tFUS neuromodulation may lead to heightened feature-based attention to visual motion stimuli.

## Methods

### Participants

Our study complies with all relevant ethical regulations regarding human research. It was reviewed and approved by the Advarra Institutional Review Board (protocol number: STUDY2017_00000426). 25 healthy human subjects (13 male / 12 female; mean age: $24.0 \pm 5.59$) were recruited from around the university area in Pittsburgh. Interested subjects were safety screened for MRI eligibility and informed of the potential risks of the study. Subjects who wished to continue gave voluntary informed consent in accordance with the World Medical Association's Declaration of Helsinki. A subject's data were excluded from analysis if their non-modulated BCI classifier was random chance. Subjects were compensated for their time at a rate of $20 per hour. We did not obtain consent to publish subject-identifying information.

### Transcranial ultrasound characterization

**Ex-vivo transcranial measurements.** A 128-element random array ultrasound transducer H275 with 700 kHz fundamental frequency (Verasonics; Kirkland, Washington, USA) was characterized in a free-water tank through a hydrated human skull fragment (Ethnicity: Caucasian, Age: 58, Gender: Male, acquired from Skulls Unlimited International, Inc.) (Fig. 6d). A hydrophone was placed on one side of the skull, opposite the ultrasound transducer. The hydrophone was moved with a stepper motor to capture the transcranial pressure field in three dimensions. Estimated target pressure, derated spatial peak pulse-average intensity ($I_{SPPA.3}$), and derated spatial peak temporal-average intensity ($I_{SPTA.3}$) were calculated to ensure safety in compliance with the current available FDA guidelines[72].

**In-silico transcranial simulations.** Computer simulations of the random array ultrasound transducer were conducted using Sim4Life (Version 7.0, Zurich Med Tech; Zurich, Switzerland). The transducer was custom-modeled and coded to emulate the H275. Subject structural MRI files were converted to pseudo computed tomography (pCT) files for use in the simulations[25,104]. The pCT images were further segmented into brain tissue and skull bone, with each assigned specific mechanical properties, such as mass density, speed of sound, and attenuation coefficients, referenced from the IT'IS 4.0 database (The Foundation for Research on Information Technologies in Society). The transducer was positioned over the head model to target the V5 area of the subject-specific brain model based on segmentation results from FreeSurfer[66,67]. The simulation considered 140 periods and 0.1 MPa transmission pressure from each element and was solved using the linear pressure wave equation (LAPWE) model.

### Neuronavigation

Each participant underwent a structure MRI at the CMU-Pitt BRIDGE Center (RRID:SCR_023356), and the acquired brain images were further segmented into brain regions using FreeSurfer[66,67]. The participant's MRI files and specific RAS (Right, Anterior, Superior) coordinates of the center of the segmented left hemisphere V5 were plugged into the Localite TMS navigator[105]. The physical size and orientation of H275 were calibrated to the navigation system and aimed at the participant's left V5.

### Online mVEP BCI speller

Each participant's head size was measured and was fit with a 64-channel EEG BrainCap TMS (Brain Products; Gilching, Germany). Incisions were made to the EEG cap to expose the V5 area, which allowed for direct interfacing between the scalp and the ultrasound transducer. Electrode placement relative to subject landmarks was captured for source imaging analysis. Raw EEG was acquired using BrainAmp (Brain Products; Gilching, Germany) at 1000 Hz. The electrode impedances were kept below 10 kΩ using conductive electrolyte gel and applied using cotton swabs. The experiment was conducted inside an acoustic and electromagnetically shielded booth (IAC Acoustics; Naperville, Illinois, USA).

The mVEP BCI speller was made in PsychoPy[106] and designed based on previously published mVEP BCI literature[42]. Subjects were presented on screen with a six-by-six virtual keyboard. They were instructed to stare at the specific key they wished to type as lines flashed to the right across each row and column of the keyboard. One scan epoch consisted of lines flashing across each row and column once. For future analysis, "on-target" will refer to the lines that flashed across the row and column of letter the subject was looking at (Fig. 6a). "Off-target" will refer to the rest of the lines in the epoch (Fig. 6b). Depending on the subject, tFUS was pulsed 1, 2, or 4 times during each scanning epoch. The first sonication began starting 100 ms before the visual stimulus onset, and any additional doses were spaced equidistant over the remaining duration of the epoch.

A model-training session was conducted where subjects were instructed to type the descending diagonal (*AHOV2_*). The model was created offline using Scikit-Learn's[107] Support Vector Machine to predict the subject's intended letter based on timing of differing neural activity of posterior electrodes P2, P1, P3, P4, P5, P6, P7, P8, PO3, PO4, PO7, PO8, O1, O2, TP7, and TP8. In five cases, additional electrodes CP1, CP2, CP3, CP4, C1, C3, C2, C4, FC1, FC3, FC2, FC4 were also used. The same electrode set was used for all conditions for a subject. A trial consisted of data from stimulus onset to 500 ms after. Data were decimated to 60 Hz, bandpass filtered from 1 to 20 Hz, and *z*-scored with respect to each channel (Fig. 6c).

For the online testing session, 16 subjects were asked to type 14 letters (*CARNEGIEMELLON*) with active SCRIBE. The other nine subjects were asked to type 15 letters (*CARNEGIE_MELLON*) without SCRIBE. Online EEG data were processed and fed to the classifier once per scan epoch. For each epoch, the classifier predicted the maximally probable row and column index of the subject's gaze. If there were multiple epoch scans per letter, the classifier would average the probabilities over the epochs.

This experiment was designed as a cross-over study, with the same subjects tested in four conditions in a predominantly randomized order: "non-modulated" (inactive tFUS), "decoupled-sham" tFUS (decoupled, but active, tFUS to account for audio-induced confounds), "US-control" (active tFUS targeted to a control brain location located by more than 1-cm away), and V5-targeted "tFUS" (estimated peak-to-peak pressure: 0.2 MPa, approximate focal beam diameter: 2 mm, approximate focal beam length: 2 mm (Fig. 6d), pulse-repetition frequency: 3 kHz, pulse duration: 200 µs, sonication duration: 500 ms (Fig. 6e). In some cases, the designated 3-hour experiment time did not allow for testing all four conditions and subjects were unable to be scheduled back. When it was apparent that time would become an issue, we prioritized testing tFUS-GC and Non-modulated conditions, given that this project was majorly to test whether tFUS neuromodulation could enhance BCI performance. The partial data for these sessions were included in the analysis if they did not meet the other exclusion criteria.

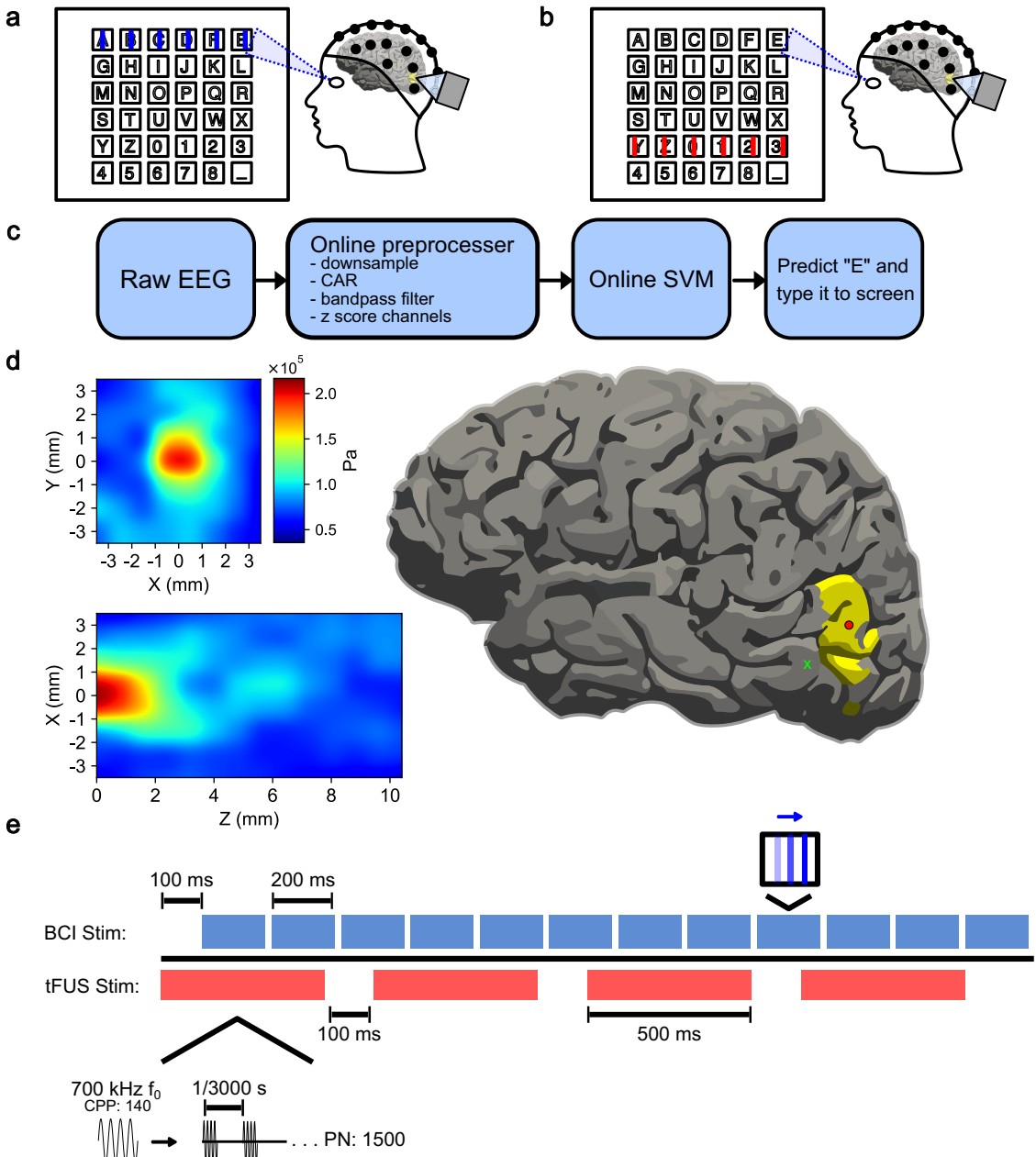

**Fig. 6 | Experimental paradigm for mVEP speller. a** Example of "on-target" visual stimuli. The lines flash across the row or column of the letter the subject is looking at. **b** Example of "off-target" visual stimuli. The lines flash on a row or column not associated with the letter of interest. **c** The online BCI decoder pipeline. The raw EEG is down-sampled to 100 Hz, bandpass filtered between 1 and 20 Hz, common average referenced (CAR), and z-scored across channels. The processed data is fed to a Support Vector Machine (SVM) classifier, predicting the letter of intent based on neural activity for on and off-target stimuli. **d** (left) The 128-element 700 kHz fundamental frequency ($f_0$) tFUS pressure wave profile measured through a skull fragment in a water tank. (right) A visualization of the profiled 2 mm focal point scaled to a 93 mm brain and centered on V5 (red circle), as well as at an example tFUS-GP area 1.41 cm away (green x). **e** A typical trial epoch consists of 12 row/column 200 ms line flashes and four 500-ms tFUS sonications. The first sonication begins 100 ms prior to the first BCI stimulation. The ultrasound parameters used for this experiment were as follows: a customized 128-element 700 kHz $f_0$, pulsed for 140 cycles (cycles-per-pulse, CPP). Each sonication consists of 1500 pulses (pulse number, PN), repeated at a pulse-repetition frequency (PRF) of 3 kHz.

Given that we did prioritize the testing of tFUS-GC and Non-modulated conditions over the other two controls, the experiments may not be considered truly 100% randomized. We account for this, as best we can, in our statistical analysis (see the Linear mixed-effect models subsection of Statistical Analysis for details).

**BCI performance analysis**
BCI performance was evaluated for each condition's online testing session based on the Euclidean distance error (EE) from the subject's intended letter to the BCI classifier's output. The output for each letter (i.e., *C, A*, etc) was considered as one separate trial. For each subject, outlier trials were identified using the interquartile range test and discarded. Afterward, all the trials for all the subjects were pooled together. The Euclidean distance was converted to a percent error by normalizing it against the maximum possible distance (diagonal distance of the grid: $6\sqrt{2}$). To account for repeated measures across subjects and the potential effects of learning and/or fatigue resulting from the number of repeated BCI scans per epoch and the order in

which the condition was tested, the Euclidean errors were fitted with a linear mixed-effect model (Eqs. 1–2). The model's fit for Euclidean errors was compared across conditions using a type III Analysis of Variance (ANOVA) test and a one-tailed $z$-test. Bonferroni $p$-value adjustment was applied to reduce the likelihood of false positive errors. Euclidean errors were considered significantly different if $p_{adjusted} < \alpha = 0.05$.

$$EE \sim Condition + Scans + Order + Order : Scans + (1|Subject) \quad (1)$$

Mathematically, this is equivalent to

$$EE_{i,k} = \beta_0 + \beta_{1j}*Condition_j + \beta_2*Scans_i + \beta_3*Order_j \\ + \beta_4*(Scans_i*Order_j) + b_{k,i} + \epsilon_{i,k} \quad (2)$$

Where $i$ corresponds to the $i$th trial of subject $k$, and $j$ denotes the experimental condition of trial $i$. $\boldsymbol{\beta_1}$ is a (1 x J) vector of the weights for each condition, and **Condition** is a (J x 1) one-hot vector. $\beta_0$, $\beta_2$, $\beta_3$ and $\beta_4$ are scalar quantities. $Scans_i$ and $Order_j$ are discrete integer values corresponding to the number of repeat scans of trial $i$ and the order in which condition $j$ was tested, respectively. $b_{k,i}$ is the random-effect of the subject $k$ for trial $i$, defined as a Gaussian distribution with a subject-specific mean and variance. $\epsilon_{i,k}$ is the error term.

## EEG preprocessing

EEG was preprocessed in Python[108] (version 3.10.9) using the MNE Python[64,109] package. Data were bandpass filtered between 1 to 40 Hz, common average referenced, and down-sampled to 100 Hz. Independent component analysis (ICA) was performed to remove eyeblink-related artifacts using MNE Python's automatic EOG detection[110], considering FP1 as an EOG channel with a $z$-threshold of 2.0. Bad epochs were rejected and sensors were cleaned using the autoreject package[111,112], and persisting eyeblink ICA bases were rejected with an additional run of MNE's automatic EOG detection using both FP1 and FP2 and a $z$-threshold of 3.5.

## EEG source imaging

Time series current density data for the left hemisphere V5 region was extracted from the preprocessed EEG data using MNE Python in the FreeSurfer[66,67] segmented region (Brodmann Area (*BA_exvivo*) atlas' *MT_exvivo-lh* label). MRI and EEG digitization data were aligned using MNE Python's automatic co-registration function[113]. The source imaging problem was solved with the Minimum Norm Estimation method[83,109]. In five cases, a subject's MRI had mesh errors that did not allow for source imaging. In these scenarios, MNE Python's base head model was used instead of the subject-specific MRI model. The source imaged time series were bandpass filtered from 1 to 40 Hz and $z$-scored with respect to each trial's 200 ms pre-stimulus baseline.

Source imaged time series were transformed into time-frequency power responses using Morlet waveforms[114]. The N200 theta, alpha, beta, and low gamma powers were extracted by taking the [4, 8), [8, 12), [12, 30), or [30, 40) Hz frequency power response, respectively, in the 100 to 250 ms post-stimulus window. Outlier trials were rejected by the interquartile range test. The data were exported to R and fit with a linear mixed-effect model to account for repeated measures across subjects (Eq. 3). To understand how this equation looks mathematically, please refer to the conversion from Equation 1 to Eq. 2. The effects of conditions were compared with a one-tailed $z$-test with false-discovery rate correction for multiple comparisons. Effects were considered significant if the adjusted $p$-value was below 0.05.

$$N200\ Power \sim Condition + (1|Subject) \quad (3)$$

This process was repeated with the Desikan-Killiany (*aparc*) atlas' parcellations for the left hemisphere superior parietal lobe

(*superiorparietal-lh*) and the left hemisphere inferior temporal lobe (*inferiortemporal-lh*) to examine possible downstream effects in visual processing pathways.

## Connectivity analysis

Cortical functional connectivity was assessed using the absolute value of Pearson's $r$ correlation coefficient. The current density time series at V1, V5, the superior parietal lobe, and the inferior temporal gyrus were calculated for each subject in each scan. These time series were then bandpass filtered 1 to 40 Hz, $z$-scored with respect to the 200 ms baseline, and subsequently filtered from 4 to 8 Hz to extract the theta band component or from 8 to 12 Hz to extract the alpha component. A correlation coefficient was calculated for each connection from epoch onset (−200 ms before stimulus) to epoch offset (500 ms post-stimulus) in the tFUS-GC, tFUS-GP, decoupled-sham, and non-modulated conditions. Subject-specific outliers were identified using the interquartile range test and removed. Correlation coefficients in the V5-IT and V5-SP were fit with a linear mixed-effect model (Eq. 4), and changes across conditions were assessed with one-tailed $z$-tests, with false-discovery rate correction for multiple comparisons. To understand how this equation looks mathematically, one can also refer to the conversion from Equation 1 to Eq. 2.

$$Pearson's\ r \sim Condition + (1|Subject) \quad (4)$$

## Statistical analysis

**Data sources.** Data for the BCI behavioral outcome came from the Euclidean distance errors of each subject's intended letters compared to the actual BCI classifier's prediction. This led to each subject having 14 (with SCRIBE target string = *CARNEGIEMELLON*), or 15 (without SCRIBE, target string = *CARNEGIE_MELLON*) datapoints per experimental condition.

For the neural analysis, since the BCI testing error was nonzero, it was not always clear whether the subjects were looking at the keys they were intended to. Therefore, to maximize the likelihood of properly matching the visual stimuli to on-target or off-target labels, electrophysiological signal analysis was performed on the EEG data collected during the BCI training session. Each on-target stimulus in a scanning epoch was considered as one trial, so each subject had 2 (one for the row index and one for the column index) * the scan repeat number * 6 (length of training set string) number of trials per experimental condition.

**EEG sensor spatiotemporal analysis.** Sensor-domain-level data for the tFUS-GC condition were analyzed against decoupled-sham, non-modulated conditions, and tFUS-GP using MNE Python's non-parametric permutation cluster tests[115]. For this, a repeated-measures $F$-test was conducted across all time and space to compare the four tested conditions, and significantly different ($\alpha = 0.05$) points that were adjacent in time and space were clustered together. Then, data across conditions were shuffled into random permutations of the original conditions. The repeated-measures F-test was run for the values at all time points and electrodes. Significant ($\alpha = 0.05$) adjacent time points and electrodes were clustered together. The number of datapoints in the largest cluster was stored and used to generate a cluster-size distribution. The cluster-forming size threshold was calculated as the 95th percentile ($p < 0.05$) from the distribution. Clusters from the original data comparison that were larger than the threshold size were deemed significant spatiotemporal clusters. This test used 1000 permutations. Significant clusters that did not include posterior electrodes or took place prior to stimulus onset were omitted. Due to the technical requirements of MNE Python's repeated measures ANOVA permutation cluster test, only data from the subjects who were tested with all four conditions ($N = 13$) were considered for this test.

**Outlier rejection.** Outliers were identified and discarded using the interquartile range method (Eq. 5). The interquartile range (IQR) is the difference between 75th percentile (Q3) and 25th percentile (Q1) of the data, and data are considered outliers if they fall 1.5 * IQR below Q1 or 1.5 * IQR above Q3. IQR rejection has been found to control for outliers even in the case of skewed data[116], and is widely used in biomedical sciences with non-normally and asymmetric distributed data[117,118].

$$IQR = Q3 - Q1$$
$$Q1 - 1.5 * IQR \leq nonoutliers \leq Q3 + 1.5 * IQR \quad (5)$$

To verify our results, we re-ran all of the tests for our major results using the double median absolute deviation (double MAD)[119] outlier rejection strategy. Data were considered outliers if they were beyond 3.5 double MAD from the median. All the statistical outcomes were validated as consistent (Supplemental Table S2).

**Linear mixed-effect models.** We analyzed the bulk of our data with linear mixed-effect models. These account for both fixed effects (i.e., experimental condition) and random effects (i.e., subject to subject differences) and allow for analysis involving repeated measures[120,121]. Tests were conducted in R[122] (version 4.3.2) with the lme4[123] and multcomp[124] packages. The resulting model residuals were non-normal; however, there are multiple studies that demonstrate the robustness of these models to residual non-normality[125,126]. This is likely because the model analysis is conducted on the model coefficients, not the raw data themselves, which are highly constrained by the nature of the models. Still, to verify the robustness of our model fits, we performed case bootstrap analysis with 1000 samples for each of our models[127], and the resulting confidence intervals are consistent with the results of the paper (Supplemental Table S3).

Data for each statistical test underwent interquartile range outlier rejection in Python, were further imported into R and fit with their specified equations (Eqs. 1–4). The post outlier rejection sample sizes are all noted on their relevant figure panels.

To conduct z-tests comparing each of the $J$ experimental conditions (in our case, $J = 4$), linear mixed-effect models produce estimates of the true mean $\boldsymbol{\beta}$ (a $J$ x 1 vector) and covariance $\boldsymbol{\Sigma}$ (a $J$ x $J$ matrix) of the experimental conditions as a multivariate mean $\hat{\boldsymbol{\beta}}$ (a $J$ x 1 vector; Eq. 6) with some covariance $\hat{\boldsymbol{\Sigma}}$ (a $J$ x $J$ matrix). These estimations are performed with R's lme4[123] function *lmer*. These values are then used by R's multcomp[124] function glht to compute z-statistics using Tukey's all pair-wise comparisons. For an in-depth description, please refer to Hothorn et al.[128]. To get the pooled standard error of any two conditions $l$ and $m$ ($SE_{lm}$), the square root of the variance of condition $l$ ($\hat{\Sigma}_{ll}$) plus the variance of condition $m$ ($\hat{\Sigma}_{mm}$) minus two times the covariance of conditions $lm$ ($\hat{\Sigma}_{lm}$) are calculated (Eq. 7). The estimated means for condition and $l$ ($\hat{\beta}_l$) and $m$ ($\hat{\beta}_m$), along with the pooled standard error, are used to compute a paired z-test statistic (Eq. 8). Multiple comparison corrections was conducted as noted under each of the relevant sections.

$$\hat{\boldsymbol{\beta}} \sim N(\boldsymbol{\beta}, \boldsymbol{\Sigma}) \quad (6)$$

$$SE_{lm} = \sqrt{\hat{\boldsymbol{\Sigma}}_{ll} + \hat{\boldsymbol{\Sigma}}_{mm} - 2 * \hat{\boldsymbol{\Sigma}}_{lm}} \quad (7)$$

$$z_{lm} = \frac{\hat{\beta}_l - \hat{\beta}_m}{SE_{lm}} \quad (8)$$

Due to previous tFUS research demonstrating ultrasound neuromodulation to enhance visual motion perception[51] and amplify event-related potential amplitudes[50,129], we had a priori hypotheses that tFUS-GC would enhance the visual-motion-based BCI performance, and that

it would amplify the relevant neural signals. As a result, we used one-tailed z-tests in our hypothesis testing for a lower Type II error rate.

We also account for the only predominantly randomized condition in these models, by starting with including a factor in them for the order in which the condition was tested (see Eqs. 1–2). This term was not a significant factor in the neural analysis models (based on a Type III ANOVA), so it was omitted.

**Cohen's *d*.** To quantify the effect size, we calculated Cohen's *d* (Eq. 9) based on the fit linear mixed-effect models. R's general linear hypothesis testing (glht) function directly returns the estimated difference in means ($\mu_1 - \mu_2$) and the pooled standard error of the means (SE) for each condition-condition comparison. It is worth noting that these values are not calculated directly from the raw data of the experiment, which, as mentioned above, are also impacted by repeated measures and additional factors like the scan number and order of testing, but instead from the linear mixed-effect model's estimates of the true effect on the mean and standard error for each condition. To convert the SE to pooled standard deviation ($\sigma_{pooled}$), we multiplied the SE by the square root of the subject number.

$$Cohen's\ d = \frac{|\mu_1 - \mu_2|}{\sigma_{pooled}} \quad (9)$$

**Bayes factor.** Another way to quantify the impact of experimental conditions on BCI Euclidean error is with Bayes Factor (BF) analysis. This analysis compares how much predictive power a model has with an additional factor (in our case, experimental condition), compared to the prior information (not knowing experimental condition). We conducted BF analysis in R using the BayesFactor package's[61] repeated measures ANOVA function (BFanova). The BFanova function cannot handle continuous data types, so our equation modeling posterior is simply Euclidean error as a function of experimental condition and subjects, with subjects being the repeated factor (Eq. 10). The prior model (Eq. 11) does not have any information about the experimental condition, and therefore only models Euclidean error as a function of the subject, with the subject being a random factor.

$$EE \sim Condition + Subject \quad (10)$$

$$EE \sim Subject \quad (11)$$

If the BF > 3, the experimental condition information added to the model (Eq. 10) is considered to have a moderate effect[62]. If the BF > 10, the information added is considered to have a strong effect. BFs are sensitive to the prior effect scaling factor[63]. Typically, only a singular value is reported, but, for a more wholistic and robust analysis[63], we present all BF for effect scaling up to BayesFactor's "ultrawide" (scaling = 1.0). Following Keysers et al.'s[62] recommendation, we ran the simulation at each effect scaling factor 1000 times and presented the median value and 95% confidence interval (Fig. 1c).

**S**hared **C**ontrol auto**R**egressive **I**ntegrated **B**ayesian **E**stimator (SCRIBE)
In order to further improve the mVEP speller classifier, a custom autocorrect program was created. This program was based on the principles of Bayesian inference, such that:

$$P(\mathbf{A}|\mathbf{B}) = \frac{P(\mathbf{A}) * P(\mathbf{B}|\mathbf{A})}{P(\mathbf{B})} \quad (12)$$

in which **A** is the corrected word, and **B** comprises the typed letters. The probability of the *corrected word* was estimated by the square-root log-frequency of the word's occurrence based on Google N Gram's

2018 database[130]. The probability of the typed letters given the corrected word was defined as 1 minus the Euclidean error between the two. This provides a similarity score between 1 (perfect match) and 0 (maximum possible error) between the typed letters and correct words, which is deemed as a proxy of the conditional probability. The probability of typed letters occurring would be the same for all possible corrected words. Thus, we can simplify the calculation by stating that conditional probability is proportional to the numerator of the expression:

$$P(\mathbf{A}|\mathbf{B}) \propto P(\mathbf{A})*P(\mathbf{B}|\mathbf{A}) \tag{13}$$

in which P($\mathbf{B}$ | $\mathbf{A}$) is the probability of the typed letters given the corrected word, and P($\mathbf{A}$) is the probability of the corrected word. This can be taken a step further, as language follows certain patterns, and some words become more probable following others. If the current word is following at least one more word, the formula changes to incorporate the past words:

$$P(\mathbf{A}|\mathbf{B},\mathbf{C}) = \frac{P(\mathbf{A})*P(\mathbf{B}|\mathbf{A})*P(\mathbf{C}|\mathbf{A},\mathbf{B})}{P(\mathbf{B},\mathbf{C})} \tag{14}$$

in which $\mathbf{C}$ comprises the previous word(s). We can make the assumption that the previous word is independent of the current letters, leading to:

$$P(\mathbf{A}|\mathbf{B},\mathbf{C}) = \frac{P(\mathbf{A})*P(\mathbf{B}|\mathbf{A})*P(\mathbf{C}|\mathbf{A})}{P(\mathbf{B},\mathbf{C})}$$
$$= \frac{P(\mathbf{A})*P(\mathbf{B}|\mathbf{A})*\frac{P(\mathbf{C})*P(\mathbf{A}|\mathbf{C})}{P(\mathbf{A})}}{P(\mathbf{B},\mathbf{C})} \tag{15}$$
$$= \frac{P(\mathbf{B}|\mathbf{A})*P(\mathbf{C})*P(\mathbf{A}|\mathbf{C})}{P(\mathbf{B},\mathbf{C})}$$

For every possible corrected word, the probability of currently typed letters occurring (P($\mathbf{B}$)), the probability of the previous word(s) occurring (P($\mathbf{C}$)), and the joint probability of the two (P($\mathbf{B}, \mathbf{C}$)) are the same. Therefore, to increase computation efficiency, these terms can be ignored in the calculation. The new equation becomes:

$$P(\mathbf{A}|\mathbf{B},\mathbf{C}) \propto P(\mathbf{B}|\mathbf{A})*P(\mathbf{A}|\mathbf{C}) \tag{16}$$

in which P($\mathbf{B}$ | $\mathbf{A}$) is the probability of the typed letters given the corrected word, and P($\mathbf{A}$ | $\mathbf{C}$) is the probability of the corrected word given the previous word(s). The probability of the corrected word occurring given previous word(s) was estimated by the scaled log-frequency of the singular words and word sequences from data in Google N Gram's 2018 multiword database.

This was tested during the online BCI session as a greedy search (Sup. Vid. S1, Sup. Vid. S2). Probabilities were calculated for all words in the Scrabble dictionary, as well as "Carnegie" and "Mellon". Three possible options were given to the subject at the end of every word. These options were the original letters typed and the next two most probable real words. This means that if the user happened to type a real word, they would be provided with the three most probable word options.

### Reporting summary
Further information on research design is available in the Nature Portfolio Reporting Summary linked to this article.

## Data availability
All data supporting the findings of this study are available within the article and its supplementary files. Any additional requests for information can be directed to, and will be fulfilled by, the corresponding authors. Source data are provided in this paper. The EEG data and Online BCI results generated in this study have been deposited in a publicly available FigShare [https://doi.org/10.6084/m9.figshare.25583334][131]. Source data are provided in this paper.

## Code availability
The R and Python analysis scripts generated and used for this study are publicly available, without restriction, on GitHub [https://github.com/bfinl/tFUS-mVEPBCI-Analysis][132].

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

## Acknowledgements

This work was supported in part by NIH grants R01NS124564 (PI: B.H.), R01AT009263 (PI: B.H.), U18EB029354 (PI: B.H.), T32EB029365 (J.K.; PI: B.H.), RF1NS131069 (PI: B.H.), R01NS096761 (PI B.H.), and R01NS127849-01A1 (PI: B.H.), as well as National Science Foundation Graduate Research Fellowship Program grant DGE2140739 (J.K.). Any opinions, findings, and conclusions, or recommendations expressed in this material are those of the author(s) and do not necessarily reflect the views of the National Institutes of Health or the National Science Foundation. The authors would also like to thank Dr. Robert Kass for advice on statistical analysis, and Dylan Forenzo for useful discussions on statistical analysis, Jenn Shanahan for assistance in EEG capping, Yunruo Ni and Chih-Yu Yeh for assistance in EEG capping and 3D printing the tFUS transducer holder, and Zherui Li for assistance in converting subject MRIs into pseudo-CT images.

## Author contributions

Conceptualization: J.K., K.Y., and B.H. Methodology: J.K., K.Y., and C.L. Data collection: J.K. and K.Y. Formal analysis: J.K. Investigation: J.K., K.Y., and B.H. Writing—original draft: J.K. Writing—reviewing and editing: J.K., K.Y. and B.H. Supervision: K.Y. and B.H.

## Competing interests

B.H. and K.Y. are co-inventors of a pending US patent application (Applicant: Carnegie Mellon University; Inventors: Bin He and Kai Yu; Application No: 18/553,901; Status: Pending; Specific aspect: tFUS and electrophysiological source imaging). J.K. and C.L. have no competing interests to declare.
