## [Peer Review File · Nature Communications]

REVIEWER COMMENTS

Reviewer #1 (Remarks to the Author):

General Comments: The manuscript explores for the first time the impact of low intensity transcranial focused ultrasonic stimulation (tFUS) to improve a brain computer interface (BCI). The authors did an impressive statistical analysis demonstrating a significant improvement of the outcome of a BCI speller despite the fact that the error reduction with tFUS (3.06%) is as low as one quarter of the standard deviation of the non-modulated measurement: the Euclidean error for the tFUS condition was $8.64 \pm 11.8\%$ as opposed to $11.7 \pm 12.8\%$ for the non-modulated and $13.8 \pm 16.9\%$ for the decoupled-sham conditions. Similarly, the statistical analysis of the EEG data (both at the sensor level and the brain region specific source level) manages to highlight changes that are four times lower than the standard deviation (see the N200 theta response with tFUS (6.98 ± 5.49 a.u.) compared to non-modulated (5.57 ± 4.16) and decoupled-sham (4.99 ± 3.60 a.u). The changes of tFUS vs US-control in theta powers in V5 are even 20 times lower than the standard deviation (lines 120-123).

Using tFUS to improve BCI control is novel and of interest. Nevertheless, the effect is statistically significant but quite weak.

Specific comments:

1. Introduction: I encourage the authors to look for pioneering tFUS papers and update their references. References are more US-centric than pioneer-centric. This is particularly true for primate work (especially for deep brain structures).
2. Introduction: "The primary brain area associated with visual motion processing is V5, or the middle temporal complex^{34,35}." It is an important claim for the choice of the TUS target, at the center of the study design. Nevertheless, reference 34 and 35 do not support the assessment. Reference 35 demonstrated a positive functional relationship between areas and areas VI /V2 and V5 during motion stimulation, reflecting only the anatomical connections between V1/V2 and V5, not that the primary brain area associated with visual motion processing is V5. Furthermore, 34 and 35 were performed in monkeys. The conclusion of reference 35 written in 1991 is that this work "promises to be a powerful tool for inferring anatomical pathways in the normal human brain." Please provide adequate reference to support the assessment in human.
3. It is stated in the methods (Online mVEP BCI speller paragraph) that the whole process " was done for four conditions in a randomized order: "non-modulated" (inactive tFUS), "decoupled-sham" tFUS (decoupled, but active, tFUS to account for audio induced confounds), "US-control" (active tFUS targeted

to a control brain location located by more than 1-cm away), and V5-targeted “tFUS”. It is indeed important to include a US-control. Figure 1 and 2 should include a full analysis including the 4 groups and not exclude the US-control.

4. As the conditions were randomized, how do the authors explain the difference between t-FUS (n=331) and US-control (n=172) (see figure 4)?

5. How do you explain the low number of decoupled-sham as well (n=229) (see figure 1)?

6. The authors are encouraged to better explain the ultrasound parameters. The parameters appear on Figure 6 but are not fully described: what does CPP, PN stand for?

7. 3D needle hydrophone scanning was performed behind only one fully hydrated real human skull sample. Where does the human sample come from? What was the age, gender, ethnicity of the donor? To what extent did it differ from the human subjects included in the study? Did the authors investigate the potential impact of different skull geometries?

8. Individual analysis of the intensity (or acoustic pressure) at target for each participant is missing. The authors used the Sim4life head model for the numerical simulations. Did the authors have access to CT or UTE images of the participants to be able to simulate the pressure distribution for each participant? Did the authors correct for the skull distortions with their multielement array? The authors need to justify their choice in the text of the article.

9. Figure 2 C: please superimpose in transparency the standard deviation curves to better assess the strength of the effect.

Reviewer #2 (Remarks to the Author):

Review ‘Transcranial Focused Ultrasound to V5 1 Enhances Human Visual Motion Brain-Computer Interface by Modulating Feature-Based Attention’ by Kosnoff et al.

This is an interesting study that addresses the effects of tFUS applied to V5 during a motion detection BCI application on accuracy of the BCI and the underlying electrophysiological changes in a group of n=21 young healthy subjects and compared it to different control conditions including no tFUS, and decoupled sham. They showed differences in Euclidian error and in theta activity between the different conditions. The authors suggest that this is the first description of neuromodulation by means of tFUS to enhance the performance of a BCI. tFUS is a novel non-invasive brain stimulation method for neuromodulation with its strength in (i) a good depth trade-off, i.e., the ability to reach deep brain structures with good topographic resolution and (ii) the possibility to combine with EEG without relevant artefacts. The present study addresses the leverage of the 2nd strength. Overall, the study is interesting and novel, however there are several important questions and significant concerns.

Intro:

- In my view the intro is somehow 'off target', the authors talk mainly about paralysis the lack of treatment and that BCI might be the solutions. Here the project is not focussed at all on motor impairment, paresis, even not on a BCI for the motor domain. Besides several of the statements in regard of motor impairment are oversimplified and even not correct. For example, the authors provide the impression that paralysis is the same independent of the underlying pathology and there are fundamental differences whether paralysis results from stroke, neurodegenerative disorders like ALS, neuro-immunological disorders like MS or peripheral problems (peripheral nerve lesions, neuro-immunological like Myasthenia). For these different disorders sharing as a symptom paralysis some of them can very well be treated and paresis will be reduced or fade away (MS, Myasthenia gravis) and some not (ALS). Also, the authors state here BCI is an approved and applied treatment e.g., in stroke rehabilitation. This is not the case, I do not think that it is FDA approved or recommended in guidelines as evidence-based treatment. Overall, the intro is an oversimplification of the problem and in my view gives the reader a wrong focus in terms of the paper. The present work does not address the problem of paralysis. Thus, the intro should significantly be revised.

- The same holds true for the abstract

Methods, Results:

Statistics, Figure 1

- Here the authors use a non-parametric Kruskal-Wallis test for independent data sets. The rationale for using a Kruskal-Wallis and not a parametric test? From the figure it looks like as if they compared individual trials (e.g., n=317 for non modulated) coming from the n=21 subjects, if I understand well. If this is the case then the individual data points are treated as independent (as they come from the n=21 subjects is not really true), however within the way of the analyses they are all treated as independent. This is in my view not correct. Maybe I misunderstood the statistical description. Comparable issues, see also stats for figure 3, 4 b

- Participants: It is not so clear, how many subjects participated in each condition and why not all in each condition, whether it is a cross over or parallel design should be clearly stated. Why was the n so different in the different conditions? Were there outliers, drop-outs?
- For some analyses only one control conditions were added to the statistical analyses. Why were not always all conditions added to the statistical modeling?
- There are different number of subjects between behavioral and the EEG analyses? What was the reason for this? Outliers? It should be described that and why there are different subject numbers
- As the control conditions are quite important, could the authors provide more details about the different control conditions?
- why did the authors different time windows for EEG-Power and EEG time frequency analyses?
- Theta is probably not the most prominent oscillations in visual processing, e.g. motion processing, there is strong evidence from e.g. the Fries lab and other that rather alpha/beta and gamma activity and their interactions are critically important. Why did the authors focus only on theta? Could they provide the results for these other important frequency bands
- If I understand well the authors provide only behavioural information about the changes in error with tFUS. Did they analyse also the speed of spelling? Did it change? It would be important whether there was a change of speed accuracy trade off with tFUS , e.g., lower errors on the price of reduced speed.
- Electrode cap: The transducer was placed over the hot spot of V5 , thus in this region there are no electrodes. I guess this indicates that it was not possible the get EEG information directly from the V5 area, but only from the surrounding electrodes? How sure are the authors that the acquired relevant brain activity representing the target brain region and not the regions around the target? This might impact on the interpretation of the findings especially the idea of inhibition and disconnection promoted, as there are known phenomena like surround inhibition (e.g., Thirugnanasambandam et al. 2015) which are not necessarily associated to the tFUS stimulation.
- ‘...Depending on the subject, tFUS was pulsed 1, 2, or 4 times during each scanning epoch...’. Could the authors please provide some explanations, justifications for these differences?
- Lowering the multiple comparison dimensionality from multiple electrodes at various timepoints to a summed theta power over the N200 time window allowed for direct comparisons across conditions with Kruskal-Wallis and Wilcoxon ranked sum testing ☑ how about for alpha or gamma power as these frequencies are also strongly involved in V5 processing
- One great strength of tFUS is the topographic resolution in the mm range. The authors defined anatomically the area for stimulation, however V5 is larger then the beam of tFUS, thus how exactly was then the right spot defined? Would it not have been better to define functionally based on neuroimaging or neurophysiological measures, e.g., based on the source analysis? With just an anatomical definition does it not lead to a loss of the topographic resolution advantage of tFUS?
- did the authors see any cross- frequency interactions modulated by tUS? This would strengthen the claims of the paper

Discussion:

- Mechanisms of action: the authors argue that ‘...Correlation analysis supports the V5-superior parietal lobe information flow is maintained from the non-modulated to the tFUS condition, while extraneous pathways are disconnected. This, contextualized with previous theta-band research^{45–50}, indicates that V5-targeted tFUS’ mechanism of action is to increase visual motion feature-based awareness....’ Here the authors argue strongly with inhibitory effects of oscillatory activity at least the cited literature indicates this. However, typical inhibitory activity is rather in the alpha frequency band and not in the theta band, furthermore the cited literature seems not be fitting well here, as this literature addresses other cognitive functions, brain areas and frequency bands.

- Would the V5 and V1 interaction described by the authors not rather be expected in such a task in the alpha/beta and gamma band rather than in the theta band (see e.g. work from the Fries Lab).

- in the discussion section the hypotheses of the authors of the underlying mechanism become not very clear, they suggest impact in inhibitory activity, which is in the occipital cortex probably more implemented by alpha then by theta activity, then they refer to changes in attention or with cognitive awareness or as results of functional disconnection aspects. Overall, it does not become clear what is the anticipated underlying mechanism, but rather gives the feeling of a patchwork explanation.

- The discussion would also profit of a short discussion of the basic underlying mechanisms of tFUS for neuromodulation, as they are under discussion. This knowledge should be discussed also in the view of the present findings.

- The tFUS impacted on behavioral performance and the EEG changes also were associated with the tFUS condition. How can the authors exclude that the respective EEG findings are rather due to the changes in behavior then really changes of brain activity induced by tFUS with following respective behavioral changes. This is always in simultaneous behavioral and neuromodulation experiment not easy to detangle, however important to know in terms of the relevance and significance of tFUS and the reported findings.

- Could the authors comment on how they think about whether the idea of supporting BCI performance by tFUS has a lot of potential. The strength of BCI’s should be to be used ideally to perform daily life activities and even self-applied by the patients. Thus, the technology for it should be easy to apply, however for tFUS the application is rather complex, especially in terms of targeting (e.g., slight change in angle of the transducer will lead to a large difference in the targeted region in the brain), the safety profile more challenging then for another noninvasive brain stimulation methods (theoretically tFUS can lead to a lesion) and it is relatively expensive. Could the authors comment on this and how they think this technology can be rally clinically translated?

- The authors give in the intro as a strong goal to provide neurotechnology to support BCIs for patients, such as stroke patients. The envisaged patients are usually in higher age. However, the full focus of the study is on healthy young subjects, so I guess a potential strong claim towards clinical applications in patient cohorts such as after stroke is still far away (lesioned brains, older age, cognitive impairment....). In terms of clinical translation, it would have been better to perform this experiment in rather healthy older (in the age of patients) or provide further evidence in patients.

Response to Reviewers

We express our sincere gratitude to the editors and reviewers for their constructive comments. In response, we have undertaken substantial efforts, including additional human experiments, computer simulations, and data analyses, and substantially revising the manuscript to address the reviewers' comments. Specifically, we conducted four more human experiments, performed thorough statistical analyses on behaviors and associated electrophysiology data, and conducted extensive transcranial focused ultrasound computer simulations using subject-specific skull models. We believe these efforts have significantly enhanced and improved the work.

Below, you will find our detailed responses to each reviewer comment in **BLUE**, with the original reviewer comments in **BLACK**. Figures in the main manuscript are denoted as Fig. x, while figures in the supplemental information are denoted as Fig. Sx. Revised or added text in the manuscript is highlighted in **RED**.

Reviewer #1 (Remarks to the Author):

General Comments: The manuscript explores for the first time the impact of low intensity transcranial focused ultrasonic stimulation (tFUS) to improve a brain computer interface (BCI). The authors did an impressive statistical analysis demonstrating a significant improvement of the outcome of a BCI speller despite the fact that the error reduction with tFUS (3.06%) is as low as one quarter of the standard deviation of the non-modulated measurement: the Euclidean error for the tFUS condition was $8.64 \pm 11.8\%$ as opposed to $11.7 \pm 12.8\%$ for the non-modulated and $13.8 \pm 16.9\%$ for the decoupled-sham conditions. Similarly, the statistical analysis of the EEG data (both at the sensor level and the brain region specific source level) manages to highlight changes that are four times lower than the standard deviation (see the N200 theta response with tFUS (6.98 ± 5.49 a.u.) compared to non-modulated (5.57 ± 4.16) and decoupled-sham (4.99 ± 3.60 a.u). The changes of tFUS vs US-control in theta powers in V5 are even 20 times lower than the standard deviation (lines 120-123). Using tFUS to improve BCI control is novel and of interest. Nevertheless, the effect is statistically significant but quite weak.

Response: Thank you for your overall favorable assessment of the novelty of our work and your meticulous observation of the results. The reported means and standard deviations encompassed the performances and neural data of all subjects, contributing to large standard deviations due to significant variations across individuals. In response to both your comment and that of Reviewer 2, we have revised our approach. In this revision, we applied a mixed effect model [1] to better accommodate the pooling of data from all subjects. Additionally, we incorporated considerations for additional factors such as fatigue/learning, which were modeled by accounting for the order of conditions and the number of repeat scans in the statistical analysis.

This study investigated the effects of naïve subjects, who were using a BCI speller paradigm for the first time. Despite efforts to simplify the task (e.g., providing an on-screen indicator), participants varied in their proficiency. It's worth noting that a future study focusing on trained subjects may produce more robust statistical results. Nevertheless, we consider the statistically significant results obtained with naïve subjects valuable for sharing with the scientific community, as they demonstrate the practical applications of the BCI speller for first-time users.

We conducted multiple comparison correction testing on all the statistics, and given the presence of four conditions, this can result in a substantial correction to minimize the risk of false-positives. Despite this correction, the results remain statistically significant, underscoring the merit of tFUS

neuromodulation in enhancing BCI performance and influencing the electrical rhythms associated with the BCI task.

[1] H. Seltman, "Mixed Models," in *Experimental Design and Analysis*, Pittsburgh, PA, 2018, pp. 357–358. [Online]. Available: <https://www.stat.cmu.edu/~hseltman/309/Book/chapter15.pdf>.

Specific comments:

1. Introduction: I encourage the authors to look for pioneering tFUS papers and update their references. References are more US-centric than pioneer-centric. This is particularly true for primate work (especially for deep brain structures).

Response: Thank you for the comment. We have significantly expanded the tFUS review section in the introduction, incorporating a multitude of additional references, particularly focusing on primate studies, as suggested. This includes a specific emphasis on targeting deep brain structures.

2. Introduction: "The primary brain area associated with visual motion processing is V5, or the middle temporal complex^{34,35}." It is an important claim for the choice of the TUS target, at the center of the study design. Nevertheless, reference 34 and 35 do not support the assessment. Reference 35 demonstrated a positive functional relationship between areas VI /V2 and V5 during motion stimulation, reflecting only the anatomical connections between V1/V2 and V5, not that the primary brain area associated with visual motion processing is V5. Furthermore, 34 and 35 were performed in monkeys. The conclusion of reference 35 written in 1991 is that this work "promises to be a powerful tool for inferring anatomical pathways in the normal human brain." Please provide adequate reference to support the assessment in human.

Response: Thank you for the excellent comment. In response, we have updated the references and now cite the following works, specifically related to visual motion processing.

Riečanský, I. Extrastriate area V5 (MT) and its role in the processing of visual motion. *Cesk. Fysiol.* **53**, 17–22 (2004).

Zihl, J., von Cramon, D., Mai, N. & Schmid, C. Disturbance of movement vision after bilateral posterior brain damage. Further evidence and follow up observations. *Brain J. Neurol.* **114 (Pt 5)**, 2235–2252 (1991).

Théoret, H., Kobayashi, M., Ganis, G., Di Capua, P. & Pascual-Leone, A. Repetitive transcranial magnetic stimulation of human area MT/V5 disrupts perception and storage of the motion aftereffect. *Neuropsychologia* **40**, 2280–2287 (2002).

3. It is stated in the methods (Online mVEP BCI speller paragraph) that the whole process " was done for four conditions in a randomized order: "non-modulated" (inactive tFUS), "decoupled-sham" tFUS (decoupled, but active, tFUS to account for audio induced confounds), "US-control" (active tFUS targeted to a control brain location located by more than 1-cm away), and V5-targeted "tFUS". It is indeed important to include a US-control. Figure 1 and 2 should include a full analysis including the 4 groups and not exclude the US-control.

Response: Thank you for the excellent suggestion. In this revision, we have included all four conditions for all comparisons and visualizations. Below are the updated Fig. 1 and Fig. 2.

Figure 1. tFUS to V5 significantly improves mVEP BCI speller outcomes. Data were fit to a linear mixed effect model to account for repeated measures across subjects and additional fixed effects from learning and fatigue (Equation 1). An ANOVA and a one-tailed t-test with Bonferroni correction were run on the model's fit for each condition. The ANOVA test indicated significant differences ($p < 0.001$) between the mean Euclidean errors for each condition. tFUS sonication to the geometric center of V5 ("tFUS-GC"; N trials = 356 trials; mean error = $13.3 \pm 18.4\%$; median error = 0.0%) leads to significantly lower Euclidean error for the mVEP BCI speller compared to non-modulated (N trials = 351; mean error = $15.5 \pm 18.7\%$; median error = 11.8%), decoupled-sham (N trials = 268; mean error = $16.9 \pm 20.8\%$; median error = 11.8%), and ultrasound steered to the geometric periphery of V5 ("tFUS-GP"; N trials = 214; mean error = $17.0 \pm 18.2\%$; median error = 11.8%) conditions. No significant differences (significance level = 0.05) between the other three conditions were found. Boxplot key: centerline = median, box-ends = 25th and 75th quartile ranges, whiskers = 1.5 * quartile ranges, triangles = mean. One-tailed t-test (with bonferroni p adjustment) key: * $p_{\text{adjusted}} < 0.05$, ** $p_{\text{adjusted}} < 0.01$, *** $p_{\text{adjusted}} < 0.001$.

Figure 2. Significant differences are found between conditions in the EEG sensor domain. a) Averaged left posterior electrode responses for 1 to 40 Hz (graph column 1), theta frequencies (graph column 2), alpha frequencies (graph column 3), beta frequencies (graph column 4), and gamma frequencies (graph column 5) for tFUS-GC (top), decoupled-sham (second row), non-modulated condition (third row), and tFUS-GP (bottom). The approximate N200 and P300 waveform responses are highlighted in yellow and purple as the 100 to 250 ms and 250 to 400 ms windows, respectively. b) Topographic maps of the trial's averaged activity for 0 to 100 ms (left), 100 to 250 ms (middle) and 250 to 400 ms (right) post stimulus filtered between 1 to 40 Hz for tFUS-GC (top), decoupled-sham ("DS"; second row), non-modulated ("NM"; third row), and tFUS-GP (bottom) conditions. The topo colormap is scaled to the relative max/min response for each. c - d) Significant spatiotemporal cluster ($p < 0.05$) between the 1 to 40 Hz filtered conditions using MNE Python's nonparametric spatiotemporal cluster test with 1000 permutations (test: repeated measures ANOVA) when comparing all four conditions. (left) The F-statistics of a significant spatial cluster denoted by white circles over the electrodes. (right) The mean activity across channels and trials of the four trial conditions (± 1 standard deviation). Gray shaded regions indicate a significant ($p < 0.05$) temporal cluster corresponding to the spatial cluster.

4. As the conditions were randomized, how do the authors explain the difference between t-FUS (n=331) and US-control (n=172) (see figure 4)?

Response: Thank you for pointing this out. In each tFUS-EEG session with naïve human subjects, a time limit of no more than 3 hours is enforced to ensure subject comfort and engagement in the BCI task. This time includes EEG electrode digitization and capping with brain target planning (approximately 1.5 hours), explanation of the BCI task with a demonstration and Q&A (duration varied by subject), brain navigation and ultrasound setup (around 0.5 hour), and necessary breaks for each subject. Given these time constraints, when it became evident that testing all 4 experimental conditions within the time limit was not feasible, we prioritized testing the real tFUS condition (tFUS-GC in this revision), the "standard" BCI (non-modulated), and one

of the two tFUS control conditions (decoupled-sham or US-control, tFUS-GP in this revision). These conditions were performed in a randomized order, resulting in noticeably smaller sample sizes for the two tFUS control conditions.

In response to the reviewer's concern, we recruited and tested an additional 4 subjects, and incorporated the data from all four conditions in this revision. Although this does not equalize the sample size across conditions, it substantially reduces proportional differences (e.g., tFUS-GC n = 356; tFUS-GP n = 214). The updated Fig. 1, combining results from all four conditions, is shown below.

5. How do you explain the low number of decoupled-sham as well (n=229) (see figure 1)?

Response: Please refer to our response to Comment 4. By including additional human subjects, we have increased the sample size for the decoupled-sham to 268, as depicted in the updated Fig. 1.

6. The authors are encouraged to better explain the ultrasound parameters. The parameters appear on Figure 6 but are not fully described: what does CPP, PN stand for?

Response: Thanks for catching these definition issues. The abbreviations have been spelled out and described in the corresponding Figure 5 caption in this revision. The updated Fig. 5 caption specifies the following: “The ultrasound parameters used for this study, including a custom 128-element 700 kHz fundamental frequency (f_0), pulsed for 140 cycles (cycles-per-pulse; CPP). This was repeated at a pulse-repetition frequency (PRF) of 3 kHz, which was repeated for 1500 pulses (pulse number; PN).”

7. 3D needle hydrophone scanning was performed behind only one fully hydrated real human skull sample. Where does the human sample come from? What was the age, gender, ethnicity of the donor? To what extent did it differ from the human subjects included in the study? Did the authors investigate the potential impact of different skull geometries?

Response: Thank you for the comment. The human skull sample (Ethnicity: Caucasian, Age: 58, Gender: Male) was obtained from Skulls Unlimited International, Inc. We conducted a 3D needle hydrophone scanning using this skull sample. In response to the reviewer’s concern, we extended the ultrasound simulation study for all the human subjects presented in this study by generating the pseudo-CT from subject-specific head MRI datasets. This approach allowed us to better estimate the ultrasound pressure field distribution within full skull cavities. As a result, we confirmed that the resultant focused transcranial ultrasound beam shapes from the computer simulations were consistent with those presented in our *ex-vivo* study. This suggests that subject-specific skull geometries may not significantly aberrate the focused ultrasound beam or warp the focal performance onto the targeted visual cortex (Fig. 5 and Supplementary Fig. S3).

Figure 5. Computer simulations of tFUS generated by the 128-element random array ultrasound transducer in a representative individual (Subject #5). a) The subject-specific T1-weighted structural magnetic resonance images in horizontal, coronal and sagittal planes, with the red cross indicating the geometrical center of the V5 area. b) Pseudo-CT images generated from the MRI in the same imaging planes as illustrated in a). c-d) The transcranial pressure field and focused ultrasound beam (-6dB focal pressure volume) coregistered with the subject skull and brain models. e) A lateral view of the transcranial ultrasound focus (-6dB focal pressure volume) on the surface of V5 along the yellow dashed line in d). The center of planned target used for optical-based brain navigation is depicted using a red cross, while the center of simulated ultrasound target is at the location marked with a green cross. The deflection of ultrasound focal spot is quantified with the spatial distance between these two crosses, which is measured as 2.84 mm for this subject.

Supplemental Figure S3. tFUS simulations using subject-specific pseudo-CT-based skull model on each of 25 human subjects, related to Results and Figure 5. a) At the skull-brain interface normal to the sonication incidence, deflections are quantified using Δx and Δy relative to the center of planned target (shown as a red cross), while the colored disks scattering around the center of planned target representing the centers of simulated targets on individual human subject. The subject number is labeled at the center of each disk. The center of planned target is employed for brain navigation based on choosing the geometrical center of segmented V5 cortical brain. b) A box plot showing the deflection data distribution of all 25 human subjects in terms of Δx , Δy and Euclidean distance. For Δx , mean: 0.11 mm, median: -0.25 mm, standard deviation: 1.49 mm. For Δy , mean: -0.10 mm, median: -0.54 mm, standard deviation: 1.08 mm. For Euclidean distance, mean: 1.71 mm, median: 1.69 mm, standard deviation: 0.60 mm. c) In addition to the computer simulation

on a representative individual (Subject #5) in Fig. 5, we presented the worst 6 cases of tFUS beam distortions in terms of the largest Euclidean distance from the center of planned target to the center of each simulated targets (the location of spatial peak pressure at the skull-brain interface). For an instance, the simulation on Subject #25 demonstrates the worst deflection of 3.05 mm (Euclidean distance) away from the center of planned target at V5. Overall, in these illustrated cases, the ultrasound focal beams still reside within the V5 region based on the individual anatomical segmentation.

8. Individual analysis of the intensity (or acoustic pressure) at target for each participant is missing. The authors used the Sim4life head model for the numerical simulations. Did the authors have access to CT or UTE images of the participants to be able to simulate the pressure distribution for each participant? Did the authors correct for the skull distortions with their multielement array? The authors need to justify their choice in the text of the article.

Response: We did not have access to subject CT images and did not correct for potential skull distortions with our customized H275 transducer. To address the reviewer’s concern, in this revision, we generated pseudo-CT from subject-specific head MRI datasets (Fig. 5). This allowed us to assess potential subject-specific ultrasound beam distortions due to different skull morphologies (Supplementary Fig. S3). Through extensive simulations, we found that subject-specific skull geometries may generate some minor deflection of the ultrasound focus (Supplementary Fig. S3), but did not significantly distort the focused ultrasound beam generated by H275 or alter the focal point of the beam onto the planned visual cortex (Fig. 5 and Supplementary Fig. S3). These results and justification are included in the revised manuscript.

9. Figure 2 C: please superimpose in transparency the standard deviation curves to better assess the strength of the effect.

Response: The updated Fig. 2c now includes superimposed standard deviation curves.

Figure 2. ... c) Significant spatiotemporal cluster ($p < 0.05$) between the 1 to 40 Hz filtered conditions using MNE Python’s nonparametric spatiotemporal cluster test with 1000 permutations (test: repeated measures ANOVA) when comparing all four conditions. (left) The F-statistics of a significant spatial cluster denoted by white circles over the electrodes. (right) The mean activity across channels and trials of the four trial conditions (± 1 standard deviation). Gray shaded regions indicate a significant ($p < 0.05$) temporal cluster corresponding to the spatial cluster.

Reviewer #2 (Remarks to the Author):

This is an interesting study that addresses the effects of tFUS applied to V5 during a motion detection BCI application on accuracy of the BCI and the underlying electrophysiological changes in a group of n=21 young healthy subjects and compared it to different control conditions including no tFUS, and decoupled sham. They showed differences in Euclidian error and in theta activity between the different conditions. The authors suggest that this is the first description of neuromodulation by means of tFUS to enhance the performance of a BCI. tFUS is a novel non-invasive brain stimulation method for neuromodulation with its strength in (i) a good depth trade-off, i.e., the ability to reach deep brain structures with good topographic resolution and (ii) the possibility to combine with EEG without relevant artefacts. The present study addresses the leverage of the 2nd strength. Overall, the study is interesting and novel, however there are several important questions and significant concerns.

Response: We appreciate the reviewer's overall favorable assessment. In response to the reviewer's specific comments, we have made significant revisions to the manuscript, as outlined below.

Intro:

- In my view the intro is somehow 'off target', the authors talk mainly about paralyse the lack of treatment and that BCI might be the solutions. Here the project is not focussed at all on motor impairment, paresis, even not on a BCI for the motor domain. Besides several of the statements in regard of motor impairment are oversimplified and even not correct. For example, the authors provide the impression that paralysis is the same independent of the underlying pathology and there are fundamental differences whether paralysis results form from stroke, neurodegenerative disorders like ALS, neuro-immunological disorders like MS or peripheral problems (peripheral nerve lesions, neuro-immunological like Myasthenia). For these different disorders sharing as a symptom paralysis some of them can very well be treated and paresis will be reduced or fade away (MS, Myasthenia gravis) and some not (ALS). Also, the authors state here BCI is an approved and applied treatment e.g., in stroke rehabilitation. This is not the case, I do not think that it is FDA approved or recommended in guidelines as evidence-based treatment. Overall, the intro is an oversimplification of the problem and in my view gives the reader a wrong focus in terms of the paper. The present work does not address the problem of paralysis. Thus, the intro should significantly be revised.

- The same holds true for the abstract

Response: Thank you for the excellent suggestion. In this revision, we have significantly revised the Introduction, as well as the Abstract, to shift focus towards tFUS development and applications for modulating brain's function. This provides the scientific background for our present work on tFUS-modulated BCI, while downplaying descriptions of BCI applications for paralysis.

Methods, Results:

Statistics, Figure 1

- Here the authors us a non-parametric Kruskal-Wallis test for independent data sets. The rational for using a Kruskal-Wallis and not a parametric test? From the figure it looks like as if they compared individual trials (e.g., n=317 for non modulated) coming from the n=21 subjects, if I understand well. If this is the case then the individual data points are treated as independent (as they come from the n=21 subjects is not really true), however within the way of the analyses they

are all treated as independent. This is in my view not correct. Maybe I misunderstood the statistical description. Comparable issues, see also stats for figure 3, 4 b

Response: Thank you for your comment. The rationale for us to use a non-parametric test (Kruskal-Wallis and Wilcoxon) is that the data are not normally distributed (based on Shapiro tests). This deviation from normality violates a core assumption of parametric counterparts such as ANOVA and t-test.

We also appreciate the reviewer's comment on repeated samples. We have addressed this concern in all our analyses by incorporating a linear mixed effect model, treating subjects as a random effect variable [1]. The model coefficients were subsequently analyzed using parametric t-tests and ANOVA.

[1] H. Seltman, "Mixed Models," in *Experimental Design and Analysis*, Pittsburgh, PA, 2018, pp. 357–358. [Online]. Available: <https://www.stat.cmu.edu/~hseltman/309/Book/chapter15.pdf>.

- Participants: It is not so clear, how many subjects participated in each condition and why not all in each condition, whether it is a cross over or parallel design should be clearly stated. Why was the n so different in the different conditions? Were there outliers, drop-outs?

Response: We are sorry for not being clear on these important aspects. The experimental design employed a cross-over approach, which is now explicitly stated in the updated manuscript. During the EEG capping process, some participants required more time than anticipated, and not all participants were willing to exceed the scheduled 3 hours or return for a follow-up session to complete all 4 experimental conditions. In such cases, tFUS targeting at the geometrical center of V5, the "standard" BCI (Non-modulated), and one of the tFUS control conditions were prioritized. To address these limitations, we tested and include additional data from four subjects with all four conditions. As mentioned by the reviewer, statistical outliers identified through the interquartile range test were removed for each analysis, as clearly noted in the manuscript.

- For some analyses only one control conditions were added to the statistical analyses. Why were not always all conditions added to the statistical modeling?

Response: For the Euclidean Error and N200 power analyses, we initially organized the manuscript to first explore the impact of ultrasound versus no ultrasound, and then investigate the significance of the targeted region. We appreciate the reviewer's comment, and in response, we have reframed the manuscript to focus simply on "V5 tFUS vs. all other controls". The figures and results have been adjusted accordingly. To better reflect the experimental setup, we have renamed the previous "tFUS" condition to "tFUS-GC", indicating the tFUS targeting at the geometrical center of V5. Additionally, the previous "US-control" condition is now labeled "tFUS-GP", denoting tFUS targeting at the geometrical periphery of V5 close to the IT region.

The primary goal of our paper is to demonstrate whether tFUS neuromodulation enhances brain-computer interface task performance, comparing the outcomes of tFUS-modulated conditions against the non-modulated BCI performance as a whole. We specifically aim to contrast holistic tFUS stimulation with the non-modulated condition, considering the latter as the true baseline control. In response to the reviewer's comments, we have incorporated all conditions for the correlation analysis and have discussed their implications in the updated manuscript.

- There are different number of subjects between behavioral and the EEG analyses? What was the reason for this? Outliers? It should be described that and why there are different subject

numbers

Response: An additional subject's US-Control (now denoted as tFUS-GP) data were included in the neural analysis compared to the behavioral analysis. This was due to time constraints preventing completion of the US-Control testing session for that particular subject. However, training session data (which mapped the subject's brain signals to the mVEP stimuli and was used to make the machine learning model for the task) were successfully collected and analyzed in the neural analysis, justifying the inclusion of this subject's data. Conversely, there is one less non-modulated subject's data analyzed in the neural analysis compared to the behavioral analysis. This discrepancy arose from a protocol-mandated time constraint of 3 hours per participant, ensuring comfort and engagement during BCI tasks. For this subject, the decoupled sham training session model was used for their non-modulated testing session. Consequently, we have behavioral data, but not from the corresponding training session for baseline analysis.

As the control conditions are quite important, could the authors provide more details about the different control conditions?

Response: US-Control (now denoted as tFUS-GP): In this condition, tFUS was targeted potentially outside or at the geometric periphery of V5, near IT. This was done to account for potential sonication location-specific effects of ultrasound while maintaining similar auditory confounds, if any. The location of H275 was fixed, and the ultrasound beam was electronically steered 2 cm away from the geometrical center of V5. Decoupled-Sham: In this condition, the acoustic aperture actively transmitted ultrasound, similar to the tFUS condition (now denoted as tFUS-GC). However, the acoustic aperture was decoupled from the scalp, positioned 4-6 cm away to create an air gap. This was designed to account for potential auditory-induced effects, as the active ultrasound produced a detectable "beep" noise during pulsing. Non-modulated: In this control condition, there was no active ultrasound (neither decoupled nor targeted elsewhere). This condition served as a control for a standard brain-computer interface as they are currently operated.

- why did the authors different time windows for EEG-Power and EEG time frequency analyses?

Response: Our EEG Power analysis was intentionally focused on capturing the N200 event-related potential, as it is believed to be the driving force of the BCI task. The time-frequency test, conducted as a permutation cluster test considering the entire epoch time window, was initially used as a high-level scan to identify general times and frequencies of interest.

However, given our direct comparison of N200 powers across multiple frequency bands, we have opted to remove the time-frequency cluster testing.

- Theta is probably not the most prominent oscillations in visual processing, e.g. motion processing, there is strong evidence from e.g. the Fries lab and other that rather alpha/beta and gamma activity and their interactions are critically important. Why did the authors focus only on theta? Could they provide the results for these other important frequency bands

Response: Thank you for your constructive comment. Initially, our focus was on the theta band due to the significantly different permutation cluster time-frequency test only returning significant clusters in the theta band. However, in response to the reviewer's comments, we have significantly expanded our analysis, results, and discussions to include other frequency bands as well. The updated findings are presented in Fig. 3.

Figure 3. tFUS-GC N200 modulation is carried through the dorsal pathway in theta and alpha frequencies. EEG source imaging of V5 yields a significant amplification of the N200 theta and alpha power for tFUS-GC modulation over decoupled-sham, non-modulated condition, and tFUS-GP control conditions, as well as significant damping of the decoupled-sham compared to the non-modulated condition. In beta and gamma frequencies, tFUS-GC has amplified power compared to the non-modulated and decoupled-sham, but not tFUS-GP. EEG source imaging of the superior parietal lobe shows significant N200 theta and alpha amplification in the tFUS-GC condition compared with the non-modulated, decoupled-sham, and tFUS-GP conditions. In beta and gamma frequency ranges, this amplification is only observed compared to the non-modulated. tFUS-GC is not significantly amplified compared to any of the control conditions in IT across any of the frequency ranges. Data were fit to a linear mixed effect model that treated subjects as random effect variables and analyzed with t-tests. Boxplot key: centerline = median, box-ends = 25th and 75th quartile ranges, whiskers = 1.5 * quartile ranges, triangles = mean. One-tailed t-test key false discovery rate adjustment: * $p_{\text{adjusted}} < 0.05$, ** $p_{\text{adjusted}} < 0.01$, *** $p_{\text{adjusted}} < 0.001$, **** $p_{\text{adjusted}} < 0.0001$.

- If I understand well the authors provide only behavioural information about the changes in error with tFUS. Did they analyse also the speed of spelling? Did it change? It would be important whether there was a change of speed accuracy trade off with tFUS , e.g., lower errors on the price of reduced speed.

Response: Thank you for the excellent comment. In our BCI paradigm, the scanning speed was constant, and it wasn't a variable that could be changed across conditions. (Each scanning epoch took 2.4 seconds, and the number of scanning epochs was kept constant for each subject across their conditions.) While spelling speed is an interesting direction for future investigation, we have included a brief discussion in the revised manuscript, as follows.

"Another limitation of our study is that we focused solely on the accuracy of the BCI speller output. Information transfer rate (ITR) is another popular metric used to quantify BCI speller performance, which relates to the speed of operation^{42,54,55}. In our paradigm, we used a constant speed for each subject by using the same number of scan repeats across conditions without any early stopping for highly probable classifier predictions. Due to the current BCI task design, we are unable to assess how tFUS neuromodulation may affect the speed of the spelling task. Future work may further investigate how tFUS neuromodulation of BCI spellers changes the ITR."

- Electrode cap: The transducer was placed over the hot spot of V5, thus in this region there are no electrodes. I guess this indicates that it was not possible to get EEG information directly from the V5 area, but only from the surrounding electrodes? How sure are the authors that the acquired relevant brain activity representing the target brain region and not the regions around the target? This might impact on the interpretation of the findings especially the idea of inhibition and disconnection promoted, as there are known phenomena like surround inhibition (e.g., Thirugnanasambandam et al. 2015) which are not necessarily associated to the tFUS stimulation.

Response: Thank you for the excellent comment. We reconstructed brain activities at V5 using EEG source imaging, based on signals recorded from all other 62 recording electrodes to reserve physical space for positioning ultrasound transducer. The EEG source imaging estimates brain electrical activity from a multitude of electrode recordings over the scalp, relying on the quantitative relationship between brain sources and the scalp field manifestation [1]. Brain activity at a given region is thus reflected by multiple electrodes over the scalp, not just the most adjacent electrode. What is crucial is sufficient spatial sampling [2], which allowed us to accurately estimate brain activity from the target brain region.

[1] B. He, A. Sohrabpour, E. Brown, and Z. Liu, "Electrophysiological Source Imaging: A Noninvasive Window to Brain Dynamics," *Annual Review of Biomedical Engineering*, vol. 20, pp. 171-196, 2018.

[2] M. Seeck, L. Koessler, T. Bast, F. Leijten, C. Michel, C. Baumgartner, B. He, and S. Beniczky, "The Standardized EEG Electrode Array of the IFCN," *Clinical Neurophysiology*, vol. 128, no. 10, pp. 2070-2077, 2017.

We have added a discussion on this issue in the revised manuscript, as follows.

"In power and correlation analyses, we reconstructed brain activities at V5 using EEG source imaging. This involved utilizing signals recorded from all other 62 recording electrodes to reserve physical space for positioning the ultrasound transducer over V5. EEG source imaging estimates brain electrical activity from a multitude of electrode recordings over the scalp, relying on the quantitative relationship between brain sources and the scalp field manifestation⁶¹. Brain activity at a given region is thus reflected by multiple electrodes over the scalp, not just the most adjacent electrode. What is crucial is sufficient spatial sampling⁸⁰, which allowed us to accurately estimate brain activity from the target brain region."

- ‘...Depending on the subject, tFUS was pulsed 1, 2, or 4 times during each scanning epoch....’. Could the authors please provide some explanations, justifications for these differences?

Response: Some subjects reported distraction due to the beeping noise of the ultrasound. In response, we reduced the number of sonications to help their focus on the task.

- Lowering the multiple comparison dimensionality from multiple electrodes at various timepoints to a summed theta power over the N200 time window allowed for direct comparisons across conditions with Kruskal-Wallis and Wilcoxon ranked sum testing how about for alpha or gamma power as these frequencies are also strongly involved in V5 processing

Response: In line with the reviewer’s suggestion, we have incorporated results for other frequency bands as well. The updated Figure 3, shown below, now includes the alpha and gamma power analyses.

- One great strength of tFUS is the topographic resolution in the mm range. The authors defined anatomically the area for stimulation, however V5 is larger than the beam of tFUS, thus how exactly was then the right spot defined? Would it not have been better to define functionally based on neuroimaging or neurophysiological measures, e.g., based on the source analysis? With just an anatomical definition does it not lead to a loss of the topographic resolution advantage of tFUS?

Response: Thank you for this insightful comment. To investigate the neural sources during the N200 time window (100 – 250 ms) of the non-modulated BCI condition, we conducted EEG source imaging and projected all the data to an average brain. The presented image represents the average response of cortical activation evoked by the mVEP BCI speller paradigm, with the V5 area segmented using FreeSurfer outlined in black. Notably, the V5 source aligns approximately with the geometric center of the area, corresponding to our targeting using optical-based brain navigation. We have included Supplementary Fig. S1 along with an explanation in the revised manuscript (1st paragraph of Discussion section).

Figure S1: The functional activity evoked by the mVEP BCI speller is located in the geometric center of V5, related to Methods and Results. EEG data from each subject's non-modulated training session were projected onto their cortical surface using EEG source imaging through minimum norm estimation. Data were bandpass filtered from 1 to 40 Hz, and the mean N200 activity (100 – 250 ms) was captured. Subsequently, the data were source morphed to a common FreeSurfer *FSAverage* brain¹¹⁴ and averaged across all subjects (N = 24). The resulting data indicate that the cortical activation associated with the BCI task was in the geometric center of V5, outlined in black and highlighted in blue. This alignment corresponds to the targeted region of tFUS-GC.

Note: As mentioned in an earlier Reviewer comment, due to a protocol-mandated time constraint, we were unable to conduct a full non-modulated training and testing set for one of our subjects. Consequently, we conducted a non-modulated testing set using the subject's decoupled-sham training-session model. Therefore, we have N = 24 subjects for non-modulated training session data.

- did the authors see any cross- frequency interactions modulated by tUS? This would strengthen the claims of the paper

Response: Thank you for the excellent suggestion. In this work, our focus is on within-frequency analysis in individual EEG bands power, specifically theta, alpha, beta and gamma, rather than exploring cross-frequency interactions. We have incorporated a discussion on the potential for future investigations of cross-frequency analyses of tFUS effect.

Discussion:

- Mechanisms of action: the authors argue that ‘...Correlation analysis supports the V5-superior parietal lobe information flow is maintained from the non-modulated to the tFUS condition, while extraneous pathways are disconnected. This, contextualized with previous theta-band research^{45–50}, indicates that V5-targeted tFUS’ mechanism of action is to increase visual motion feature-based awareness....’ Here the authors argue strongly with inhibitory effects of oscillatory activity at least the cited literature indicates this. However, typical inhibitory activity is rather in the alpha frequency band and not in the theta band, furthermore the cited literature seems not be fitting well here, as this literature addresses other cognitive functions, brain areas and frequency bands.

Response: Thank you for the excellent comment. In response to this and other comments from the reviewer, we have greatly expanded our analyses covering multiple frequency bands, including alpha band as presented in the updated Figures 3 and 4. In addition to amplified theta powers, we see amplified N200 alpha powers as well in the dorsal processing pathway. We have updated our discussion to attribute inhibitory processes to the heightened alpha band.

“The specific mechanisms underlying focused ultrasound neuromodulation are still being investigated^{4,84–86}. One early hypothesis suggested that tFUS increases local interneuron firing, influencing the flow of information⁴⁹. This aligns with our study’s observation of increased alpha power, as alpha oscillations are often associated with inhibitory, top-down attention processes^{69–71}.”

Figure 3. tFUS-GC N200 modulation is carried through the dorsal pathway in theta and alpha frequencies. EEG source imaging of V5 yields a significant amplification of the N200 theta and alpha power for tFUS-GC modulation over decoupled-sham, non-modulated condition, and tFUS-GP control conditions, as well as significant damping of the decoupled-sham compared to the non-modulated condition. In beta and gamma frequencies, tFUS-GC has amplified power compared to the non-modulated and decoupled-sham, but not tFUS-GP. EEG source imaging of the superior parietal lobe shows significant N200 theta and alpha amplification in the tFUS-GC condition compared with the non-modulated, decoupled-sham, and tFUS-GP conditions. In beta and gamma frequency ranges, this amplification is only observed compared to the non-modulated. tFUS-GC is not significantly amplified compared to any of the control conditions in IT across any of the frequency ranges. Data were fit to a linear mixed effect model that treated subjects are random effect variables and analyzed with t-tests. Boxplot key: centerline = median, box-ends = 25th and 75th quartile ranges, whiskers = 1.5 * quartile ranges, triangles = mean. One-tailed t-test key false discovery rate adjustment: * $p_{\text{adjusted}} < 0.05$, ** $p_{\text{adjusted}} < 0.01$, *** $p_{\text{adjusted}} < 0.001$, **** $p_{\text{adjusted}} < 0.0001$.

Figure 4. tFUS-GC's effects are carried through the theta frequency of the dorsal pathway. a) The canonical dorsal and ventral visual processing pathways with their FreeSurfer associated labels. b-e) Grand-average |Pearson's r | values for theta frequency absolute correlation for b) non-modulated, c) tFUS-GC, d) decoupled-sham, and e) tFUS-GP of the mVEP BCI epochs. f) The dorsal pathway (V5-SP) theta frequency band correlation is significantly reduced for the decoupled-sham condition compared to the others. g) The ventral pathway (V5-IT) theta frequency band is significantly decorrelated for both the tFUS-GC and decoupled-sham compared to the non-modulated condition. h) There is no significant difference across alpha frequency correlations for any of the conditions in the dorsal pathway. i) tFUS-GC and the decoupled-sham alpha frequency bands are significantly decorrelated compared to the non-modulated and tFUS-GP conditions, but not from each other. Data were fit and analyzed with a linear mixed effect model. Boxplot key: centerline = median, box-ends = 25th and 75th quartile ranges, whiskers = 1.5 * quartile ranges, triangles = mean. One-tailed t-test with false-discovery rate correction key: * $p_{\text{adjusted}} < 0.05$, ** $p_{\text{adjusted}} < 0.01$.

- Would the V5 and V1 interaction described by the authors not rather be expected in such a task in the alpha/beta and gamma band rather than in the theta band (see e.g. work from the Fries Lab).

Response: In this revision, in light of the significantly expanded analyses on V5 – IT and V5 – SP, we have elected to remove the analysis of the V5-V1 pathway, to focus to the feed-forward dorsal and ventral pathways extending from the sonication site.

With regards to the different frequency bands in in those pathways, we see significantly elevated alpha N200 power in the tFUS-GC condition compared to all three control conditions, but this is not the case for beta and gamma. We discuss this with regards to literature reporting alpha’s role in inhibitory processes.

“The specific mechanisms underlying focused ultrasound neuromodulation are still being investigated^{4,84–86}. One early hypothesis suggested that tFUS increases local interneuron firing, influencing the flow of information⁴⁹. This aligns with our study’s observation of increased alpha power, as alpha oscillations are often associated with inhibitory, top-down attention processes^{69–71}. More specifically, Payne et al. found that increased alpha activity was linked to an improved ability to ignore distracting information⁷⁰.”

We continue to see significantly elevated theta power in the tFUS-GC condition compared to the other three controls as well. We acknowledge the controversial role of occipital theta in attention-related processes, and address it by first turning our discussion to parietal theta, which is more widespread acknowledge to be increased with increased attention.

“The role of theta power in visual attention seems to be somewhat controversial, as it has been reported that occipital theta is both decreased⁸⁷ and increased⁷² with attention to visual stimuli. We address this inconsistency by first focusing on the enhanced tFUS-GC theta power in the superior parietal lobe. There is a larger consensus that increased theta power in the parietal region is tied to increased attention^{72–75}. Therefore, tFUS-GC may increase feature-based attention downstream the dorsal pathway, at the very least. It would be counterintuitive for tFUS to decrease attention-related activities at its site of sonication while only to increase it later downstream, so we view it as far more likely that the observed heightened occipital theta power is also linked to increased attention.”

- in the discussion section the hypotheses of the authors of the underlying mechanism become not very clear, they suggest impact in inhibitory activity, which is in the occipital cortex probably more implemented by alpha then by theta activity, then they refer to changes in attention or with cognitive awareness or as results of functional disconnection aspects. Overall, it does not become clear what is the anticipated underlying mechanism, but rather gives the feeling of a patchwork explanation.

Response: Thank you for the comment. After rerunning the power analysis for multiple frequency bands, we observed the same significant amplifications in the alpha band, in addition to the theta band. Our analysis has been updated to include a discussion on alpha inhibition as well. Very briefly: tFUS-GC was found to significantly increase alpha power in the dorsal pathway, which likely acts as a gating mechanism to stop additional flow of information into the dorsal pathway, and is consistent with a theory of tFUS’ mechanism of action that it works through filtering out unnecessary neuronal input [1].

Furthermore, we significantly expanded our discussion to address questions related to interactions and connectivity in other frequency bands. Briefly: we maintain our position that increased theta power is linked to increased attention. The weakening of theta-band correlation between tFUS-GC and the non-modulated condition also likely serves to disrupt attentional sampling. We have significantly expanded our discussion and citations to make this point. For the beta and gamma band N200 powers, we find no significant differences between tFUS-GC and tFUS-GP conditions.

Please see our updated discussion for the full write up with citations. We believe this revision presents a more cohesive and comprehensive explanation.

[1] W. Legon et al., "Transcranial focused ultrasound modulates the activity of primary somatosensory cortex in humans," *Nat. Neurosci.*, vol. 17, pp. 322–329, 2014.

- The discussion would also profit of a short discussion of the basic underlying mechanisms of tFUS for neuromodulation, as they are under discussion. This knowledge should be discussed also in the view of the present findings.

Response: We have added some discussions on tFUS mechanisms of action in view of our power and correlation analysis findings.

In this revision, we also added a short discussion of the basic underlying mechanism of ultrasound neuromodulation.

"The specific mechanisms underlying focused ultrasound neuromodulation are still being investigated^{4,84–86}. One early hypothesis suggested that tFUS increases local interneuron firing, influencing the flow of information⁴⁹. This aligns with our study's observation of increased alpha power, as alpha oscillations are often associated with inhibitory, top-down attention processes^{69–71}. More specifically, Payne et al. found that increased alpha activity was linked to an improved ability to ignore distracting information⁷⁰. Combining these findings supports the idea that tFUS enhances target-location specific activities by filtering out distracting information."

Due to the robustness of our control conditions, we are also able to join the discussion on the extent of the auditory effect on tFUS-induced neural activation.

"It is well accepted in the tFUS community that noise alone can induce neurological responses⁹⁰. However, researchers hold different views on how significant a role sound-based modulation plays. Some assert that the noise has no effect on the modulation¹⁵, and mitigating it does not affect the outcome on motor responses⁹¹. Others contend that the entirety of tFUS-based neuromodulation is due to auditory side effects^{92,93}. The results from our study acknowledge the target-specific tFUS neuromodulation effects and the impact of auditory-related brain responses. Furthermore, our correlation results indicate that, at least in the case of attention-based tasks, the auditory effect may cause wide-spread circuitry disconnection, with tFUS selectively reconnecting pathways based on the stimulation targets."

- The tFUS impacted on behavioral performance and the EEG changes also were associated with the tFUS condition. How can the authors exclude that the respective EEG findings are rather due to the changes in behavior than really changes of brain activity induced by tFUS with following respective behavioral changes. This is always in simultaneous behavioral and neuromodulation experiment not easy to detangle, however important to know in terms of the relevance and significance of tFUS and the reported findings.

Response: Thank you for the excellent question! Our results clearly demonstrate the impact of tFUS neuromodulation on behavioral performance, while at the same time, we monitor brain activity using simultaneous EEG recordings. It is well known that EEG reflects instantaneous brain activity. The reviewer raised an excellent question that if tFUS is able to induce brain response in electrical activity, or such EEG changes are due to various behavioral performance. In the following, we argue our EEG findings were associated with tFUS sonications, instead of the respective EEG findings are due to the changes in behavior that triggered EEG changes.

- 1) In designing the experimental paradigm, we minimized learning-based aspects of the task. We provided subjects with an indicator on the key they were supposed to be looking at, ensuring no change in behavior from a keyboard navigational standpoint. The speller itself is a rather passive BCI (subjects look at a visual stimulus, and we decode from the visual cortex), as opposed to more active control BCI (i.e. sensorimotor, motor), so there is not much to learn about the task behaviorally. Nevertheless, we randomized the order of the experiments across subjects. Even if there was a general trend of learning a new behavior, it would be equally present and averaged out across the different experimental conditions.
- 2) We conducted a separate study to demonstrate that tFUS stimulation alone, administered during the resting state, is capable of inducing brain electrical activity localized to the tFUS targeting region, as recorded from scalp EEG. Fig. R1 illustrates an example of tFUS-induced brain electrical activity in V5, estimated from EEG in a healthy human subject. Fig. R1 underscores that tFUS stimulation can directly induce brain electrical activity, detectable by EEG.

Figure R1: Example of tFUS-induced brain electrical activity at V5 in a human subject. tFUS (128 element transducer, 700 kHz f_0 , 300 Hz PRF, 0.2 MPa peak-to-peak pressure) was targeted to V5 (outlined in black; highlighted in purple) with EEG recorded 100 trials. The subject was instructed to sit still and focus on a fixated computer screen symbol while this occurred. The trial-average EEG data were bandpass projected back to their cortical surface using minimum norm estimation. Source data were filtered 1 to 50 Hz, and the mean activity from tFUS onset to 500 ms afterwards was captured (sonication duration: 500 ms). The resulting plot is for a single subject and showcases tFUS' ability to activate brain activity as the sole stimulus without visual stimulus. The induced electrical brain activity is marked by reds/yellow inside of the V5 region.

While it is challenging to disentangle in simultaneous behavioral and neuromodulation experiments, our EEG findings provide additional evidence to better understand the impact of tFUS neuromodulation.

- Could the authors comment on how they think about whether the idea of supporting BCI performance by tFUS has a lot of potential. The strength of BCI's should be to be used ideally to perform daily life activities and even self-applied by the patients. Thus, the technology for it should be easy to apply, however for tFUS the application is rather complex, especially in terms of

targeting (e.g., slight change in angle of the transducer will lead to a large difference in the targeted region in the brain), the safety profile more challenging than for another noninvasive brain stimulation methods (theoretically tFUS can lead to a lesion) and it is relatively expensive. Could the authors comment on this and how they think this technology can be really clinically translated?

Response: Thank you for the excellent question. You are correct in noting that, as it currently stands, tFUS technology is somewhat complex to use in conjunction with BCI. However, we are optimistic about the development of medical technology – we can now take EKG readings on our wristwatches! While admittedly, tFUS administration is different from reading heart signals, it is not infeasible to us that focused ultrasound technology may continue to improve in terms of compactness and usability. As this is the first experiment to demonstrate the impact of tFUS in enhancing BCI performance, what we have shown is the merit and potential for future daily applications. Regarding safety, tFUS may lead to a lesion if using high-intensity focused ultrasound. There have been a number of low-intensity tFUS human studies without reported safety concerns [1], including those tFUS neuromodulation literature on human as we cite in this revision.

[1] W. Legon, S. Adams, P. Bansal, et al., "A retrospective qualitative report of symptoms and safety from transcranial focused ultrasound for neuromodulation in humans," *Sci Rep*, vol. 10, p. 5573, 2020, <https://doi.org/10.1038/s41598-020-62265-8>.

- The authors give in the intro as a strong goal to provide neurotechnology to support BCIs for patients, such as stroke patients. The envisaged patients are usually in higher age. However, the full focus of the study is on healthy young subjects, so I guess a potential strong claim towards clinical applications in patient cohorts such as after stroke is still far away (lesioned brains, older age, cognitive impairment....). In terms of clinical translation, it would have been better to perform this experiment in rather healthy older (in the age of patients) or provide further evidence in patients.

Response: Thank you for your comment. While our study opens up avenues for enhancing BCI performance using tFUS neuromodulation, we have included a discussion in the revised manuscript, emphasizing the necessity of future investigations to directly test the clinical translation in various patient cohorts and in older subjects.

REVIEWER COMMENTS

Reviewer #1 (Remarks to the Author):

General Comments: I would first like to emphasize that the authors did an extensive work to address the reviewers' comments. Some of their additional work go beyond my expectations. Nevertheless, they rose novel and significant concerns (see specific comments). The statistical analysis is now too complex for me to evaluate and would require analysis by a statistician.

Specific comments:

1. I recommend to ask a statistician to look at the mixed effect model used in the paper and the overall effect size. The methodology lacks reference: H. Seltman is mentioned in the responses to reviewers but not in the paper. In their response to reviewer 2, the authors first state that "the data are not normally distributed (based on Shapiro Test). This deviation from normality violates a core assumption of parametric counterparts such as ANOVA and t-test". But later they state that "The model coefficients were subsequently analyzed using parametric t-tests and ANOVA" after using the linear mixed effect model. I am confused.
2. Overall, what is the effect size ? Please quantify.
3. The authors now include 4 additional volunteers with a different protocol : they all had the four trials whereas previous volunteers had three trials only. Is it legitimate to pull them all together?
4. (follow up of previous question): when did the authors notice that "Given these time constraints, [...] it became evident that testing all 4 experimental conditions within the time limit was not feasible"? When did you decide to prioritize testing three conditions in priority? What is the impact on the results? The authors mentioned that they incorporated considerations for additional factors such as fatigue and learning. What is the impact of changing the number of conditions?
5. Figure 2: the authors the mean activity across channels and trials for the four trials +/- 1 standard deviation. Did the authors report standard deviations or standard deviations of the mean? Standard deviations seem low compared to the plots that were provided in the manuscript that was previously submitted.
6. Simulations: what is the average shift in the axial (z) dimension? Only 2D shifts are reported. Also, it seems that the pressure maps are thresholded and that the maximum pressure amplitude is outside the brain. What is the ratio between the maximum pressure in the brain and the maximum pressure in the head? Where is located the maximum pressure ? The transmitted intensity in the brain is missing: please provide a table with the estimated intensity for each volunteer.

7. the authors state that the skulls were acquired from Skulls Unlimited. Did the donor give full consent? Was it approved by the IRB of Carnegie Mellon University? Boston University?

Reviewer #2 (Remarks to the Author):

The authors performed an extensive revision and response to the raised points. This allowed to clarify several aspects, but could not address all of the important aspects. The additional analyses were very helpful, but opened or raised novel critical points.

The authors changed now completely the introduction, overall I think it fits clearly better to the purpose of the study than the previous version of it. In my view it is a bit lengthy and repetitive in certain points. The authors state here that V5 is the main/primary brain area processing motion 'The primary brain area associated with visual motion processing is V5...'. I do not think that this is correct, there is significant literature showing that V1 is already involved in motion detection/processing in interaction with V5, thus the analyses of the interaction of both would be also quite important.

Overall the statistics is still not very clear or convincing for me. In e.g., Fig1 of the response letter there are still individual trials in the ANOVA included, in my view it clearly violates core assumptions of independence of an ANOVA. Furthermore, why is a one-tailed test used? I do not think that there were such strong assumption for all the conditions that one-tailed test are sufficient. Furthermore, in Fig. 3 for every frequency band different trial numbers are used? In the view of the quite small effect sizes also a Bayesian approach would probably provide a more clear picture in regard of the effects.

Subjects and conditions: the authors state that due to time restrictions they prioritized conditions, thus the present trial was not a randomized trial as stated in the MS, but the conditions were prioritized and in a non-randomized order, which might have significantly influenced the findings, as factors like drowsiness or surprise, novelty at the beginning of the task, learning might play a significant role

I still do not fully agree with the authors' selection of comparisons of conditions. I guess they had an a priori hypothesis which indicated an statistical analysis to test the primary hypothesis and I guess it included here all control conditions in the hypothesis and not only verum vs sham? Thus the argumentation in terms of the used contrasts is not so clear. This is especially important in term of the small effect size

The authors provided nicely more details about the different conditions, especially about the tFUS-GP condition. In the view of the analyses of the activity provided in Fig. S1, one gets the impression that the beam of the tFUS-GP (2cm from hotspot) is actually still in the activated area.

The authors provided further analyses in other important frequency bands for such a task e.g., the alpha, beta and gamma band (Fig. 3). From these new analyses, the effects induced by tFUS appears to be rather unspecific in terms of frequency with differences between all different conditions and not only between the verum condition and the control. This creates quite relevant doubts in terms of the physiological meaning or specificity of the physiological meaning of the tFUS impact on brain oscillatory activity in the visual system. Also here the n for every single condition and frequency band is different? What is the reason for this? Artefact rejection? Should it not be the same? Also here in the view of the multitude of posthoc testing and that the authors apparently did not have a frequency specific hypothesis in my view one-tailed testing is not an adequate approach. What results would two-tailed testing have provided?

The different pulsing might have influenced the results. Did the authors compare subjects with 1, 2, 4 times pulsing? This might have led to distraction, fatigue etc in an incoherent way?

The authors provided an informative overview of the activity in the visual cortex during the task (Fig. S1) and how the focus of the tFUS beam was. It shows that the beam is in the activation area, however, it suggests also that the tFUS-GP control condition might actually have also reached the relevantly activated areas. Furthermore, this is a group average with rather widespread activity there might be relevant differences in individual subjects, that might lead to heterogeneity . Thus it would be interesting to see individual overlaps between the sources and the tFUS beam of individual subjects.

I think it would be of large interest in the present data to look for cross-frequency interactions, especially in the view of the results of the new analyses that do show changes in all frequency bands. It would help to better understand the specificity of the tFUS effects here.

The discussion does not provide a clear explanations and concept to explain the changes in basically all conditions and all frequency bands. It is probably also very difficult to define a general conceptual framework of the tFUS effect as the behavioral effects are quite small and the electrophysiological results based on the new analyses seem to be quite unspecific.

Response to Reviewers

We thank the reviewers for their additional constructive comments. We have made substantial efforts to address all comments raised by the reviewers, including conducting additional analyses, adding a statistical analysis sub-section, updating one main figure, adding 1 supplementary figure and 2 supplementary tables. We believe the manuscript has been further enhanced and improved. Below, you will find our detailed responses to each reviewer comment in **BLUE**, with the original reviewer comments in **BLACK**. Revised or added text in the manuscript is highlighted in **RED**.

Reviewer #1 (Remarks to the Author):

General Comments: I would first like to emphasize that the authors did an extensive work to address the reviewers' comments. Some of their additional work go beyond my expectations. Nevertheless, they rose novel and significant concerns (see specific comments). The statistical analysis is now too complex for me to evaluate and would require analysis by a statistician.

Response: We thank the reviewer for acknowledging the work we put into the revisions and the overall positive sentiment. We have addressed the specific comments below. While we do not think the statistical analysis to be particularly complex, we have consulted an expert in statistics on the present statistical analysis, and would also welcome further review by an additional statistician on the matter.

Specific comments:

1. I recommend to ask a statistician to look at the mixed effect model used in the paper and the overall effect size. The methodology lacks reference: H. Seltman is mentioned in the responses to reviewers but not in the paper. In their response to reviewer 2, the authors first state that “the data are not normally distributed (based on Shapiro Test). This deviation from normality violates a core assumption of parametric counterparts such as ANOVA and t-test”. But later they state that “The model coefficients were subsequently analyzed using parametric t-tests and ANOVA” after using the linear mixed effect model. I am confused.

Response: Thank you for your excellent question, and we apologize for not clarifying this point further previously. When analyzing the linear mixed effect models, the fit model coefficients, not the raw data themselves, are what get analyzed. We have elaborated on this point in our new Statistical Methods section, as well as run bootstrap analysis to support the consistency of the results. Your comment has also caused to realize we made a typo in our manuscript – the R multcomp function we used compares models using z-tests, not t-tests, and we have fixed this typo in our manuscript.

“We analyzed the bulk of our data with linear mixed effect models. These account for both fixed effects (i.e. experimental condition) and random effects (i.e. subject to subject differences) and allow for analysis involving repeated measures^{118,119}. Tests were conducted in R¹²⁰ (version 4.3.2) with the lme4¹²¹ and multcomp¹²² packages. The resulting model residuals were non-normal; however, there are multiple studies that demonstrate the robustness of these models to residual non-normality^{123,124}. This is likely because the model analysis is conducted on the model coefficients, not the raw data themselves, which are

highly constrained by nature of the models. Still, to verify the robustness of our model fits, we performed case bootstrap analysis with 1,000 samples for each of our models¹²⁵, and the resulting confidence intervals are consistent with the results of the paper (Supplemental Table S2).”

2. Overall, what is the effect size? Please quantify.

Response: Thank you for the great question. To quantify the effect, we have calculated $|Cohen's d|$ value, which is defined as $|mean_1 - mean_2| / SD$, and included the analysis in our manuscript as Figure 1b. Below is the relevant panel and caption.

Figure 1b) The effect size was quantified using Cohen's d (Equation 4). tFUS-GC had moderate ($d \geq 0.5$ ⁶⁰) effects on BCI outcomes compared to all three control conditions

We have also referenced the results in the relevant portions of the Results and Discussion, and included the calculation details in the new Statistical Analysis methods subsection.

3. The authors now include 4 additional volunteers with a different protocol: they all had the four trials whereas previous volunteers had three trials only. Is it legitimate to pull them all together?

Response: Thank you for the excellent question. There was no difference in protocol with the new subjects. When we had mentioned that we tested all four conditions on the new subjects, that was partially brought up due to other concerns about unequal sample sizes. The experiment protocol for these new subjects was exactly the same as it was for the previous subjects who had time to complete all four experimental conditions.

We can account for the subjects that did not have time to complete all four conditions with our Linear Mixed Effect model, by including the *Scans*, *Order*, and *Order:Scans* terms where *Order* is the order in which a condition was tested, and *Scans* is the number of scan repeats per letter (a proxy for trial time). Further modeling details are presented below with regard to Question 4.

We added some short discussions in our Methods section relevant to this:

Online mVEP BCI speller

“In some cases, the approved 3-hour experiment time did not allow for testing all four conditions and subjects were unable to be scheduled back. When it was apparent that time would become an issue, we prioritized testing tFUS-GC and Non-modulated conditions, given that this project was majorly to test whether tFUS neuromodulation could enhance BCI performance. The partial data for these sessions were included in analysis if they did not meet the other exclusion criteria.

Given that we did prioritize the testing of tFUS-GC and Non-modulated conditions over the other two controls, the experiments may not be considered truly 100% randomized. We account for this, as the best we can, in our statistical analysis (see the *Linear mixed effect models* subsection of *Statistical Analysis* for details.)”

Linear Mixed Effect Models:

“We also account for the only predominantly randomized condition in these [linear mixed effect] models, by starting with including a factor in them for the order in which the condition was tested (see Equation 1). This term was not a significant factor in the neural analysis models (based on a Type III ANOVA), so it was omitted.”

4. (follow up of previous question): when did the authors notice that “Given these time constraints, [...] it became evident that testing all 4 experimental conditions within the time limit was not feasible”? When did you decide to prioritize testing three conditions in priority? What is the impact on the results? The authors mentioned that they incorporated considerations for additional factors such as fatigue and learning. What is the impact of changing the number of conditions?

Response: We prioritized testing tFUS-GC and the non-modulated BCI in all of our subjects, as our main hypothesis was to investigate whether tFUS to the center of V5 (tFUS-GC) could enhance BCI outcomes. When time permitted, we tested all four conditions, but when time did not allow for it, we did not prioritize one of the active tFUS controls over the other.

We can view the effect of changing the number of conditions by asking, “does the *order* variable explain a significant portion of result?” Adding a 4th condition should not change the results of the previously tested condition for the subject, and, vice versa not having time for a 4th test should not change what was already tested. However, testing a condition 1st vs. 4th may change the result due to fatigue/learning/etc. For the behavioral BCI metric, we do see a significant contribution of condition order (tested with a Type III ANOVA; Figure I) on the EE behavior.

Type III Analysis of Variance Table with Satterthwaite's method							
	Sum Sq	Mean Sq	NumDF	DenDF	F value	Pr(>F)	
Condition	5334.8	1778.3	3	1170.6	6.5197	0.0002254	***
Order	4021.8	4021.8	1	1176.2	14.7453	0.0001296	***
Scans	1235.8	1235.8	1	65.1	4.5309	0.0370765	*
Order:Scans	6329.6	6329.6	1	1180.8	23.2065	1.644e-06	***

Figure I: Type III ANOVA Results for Euclidean Error

However, for the neural data, the *order* did not play a significant factor in the result (Figure II). Please note that each *scan* was considered a trial of the neural data. Because of this, the number of scans per trial (*Scans*) was not a factor in the neural data model, as the scans per trial would be equal to one for all trials.

Type III Analysis of Variance Table with Satterthwaite's method							
	Sum Sq	Mean Sq	NumDF	DenDF	F value	Pr(>F)	
condition	3505.3	1168.4	3	4622.0	46.9227	<2e-16	***
order	24.2	24.2	1	4608.1	0.9718	0.3243	

Figure II: Type III ANOVA Results for N200 theta power at V5.

We were able to take this into account with our Linear Mixed Effect models by including *order* as a factor in the model for behavioral results. We omitted the term in our neural analysis due to its insignificance.

We point again to our included discussion in the Statistical Methods on the matter:

“We also account for the only predominantly randomized condition in these [linear mixed effect] models, by starting with including a factor in them for the order in which the condition was tested (see Equation 1). This term was not a significant factor in the neural analysis models (based on a Type III ANOVA), so it was omitted.”

5. Figure 2: the authors the mean activity across channels and trials for the four trials +/- 1 standard deviation. Did the authors report standard deviations or standard deviations of the mean? Standard deviations seem low compared to the plots that were provided in the manuscript that was previously submitted.

Response: The plot in Figure 2 is of the grand average of all the electrodes highlighted with white circles (i.e., the significant cluster). The plotting software, MNE Python, handles plotting of the grand average error margins in terms of confidence intervals. We plotted a 68% confidence interval, which is approximately equivalent to ± 1 standard deviation.

We are unsure which plots the reviewer is referring to with regards to the “previously submitted” manuscript. Our initial submission did not have the error included in the permutation cluster test plots and we were requested to add them. The R1 revised submission was the first with the plotted errors.

6. Simulations: what is the average shift in the axial (z) dimension? Only 2D shifts are reported. Also, it seems that the pressure maps are thresholded and that the maximum pressure amplitude is outside the brain. What is the ratio between the maximum pressure in the brain and the maximum pressure in the head? Where is located the maximum pressure? The transmitted intensity in the brain is missing: please provide a table with the estimated intensity for each volunteer.

Response: The average axial shift of transcranial ultrasound focus from the skull-brain interface is 4.65 mm (see all ΔZ in Table R1). The reviewer is right that the pressure maps are thresholded to show the -6 dB focal volume and the maximum pressure amplitude is outside the brain. The Table

R1 below lists all the ratios (*Pressure ratio*) between the maximum pressure in the brain and the maximum pressure in the head, and the mean value of those pressure ratios is 0.14. In the computer simulations, the maximum pressure is mostly located at the scalp-skull interface with 22 subjects presenting the maximum localized within the scalp layer, and 3 subjects showing the maximum localized within the skull layer. The estimations of transmitted intensity in the brain (derated spatial-peak temporal-average intensity $I_{SPTA.3}$ and derated spatial-peak pulse-average intensity $I_{SPPA.3}$) for all subjects are listed in Table R1 below.

Table R1. Subject-wise characterization of transcranial ultrasound focus

Subject ID	ΔZ (mm)	Pressure ratio	$I_{SPTA.3}$ (mW/cm ²)	$I_{SPPA.3}$ (W/cm ²)
1	1.93	0.14	185.41	0.31
2	11.26	0.08	60.78	0.10
3	2.15	0.15	221.24	0.37
4	11.56	0.11	117.52	0.20
5	6.85	0.17	260.14	0.43
6	1.90	0.17	259.74	0.43
7	0.84	0.11	115.35	0.19
8	8.54	0.08	60.20	0.10
9	7.22	0.18	302.08	0.50
10	0.70	0.06	39.23	0.07
11	2.91	0.10	89.36	0.15
12	1.75	0.24	530.16	0.88
13	1.27	0.15	224.10	0.37
14	5.77	0.14	185.00	0.31
15	10.87	0.26	619.17	1.03
16	3.24	0.25	576.74	0.96
17	4.20	0.18	292.25	0.49
18	4.97	0.12	123.67	0.21
19	1.43	0.14	187.63	0.31
20	8.41	0.15	210.97	0.35
21	10.38	0.07	48.27	0.08
22	0.57	0.14	181.40	0.30
23	1.37	0.14	197.47	0.33
24	4.09	0.11	116.30	0.19
25	2.10	0.14	197.42	0.33

Note: ΔZ refers to axial shift of transcranial ultrasound focus (maximal ultrasound pressure magnitude in the brain) from the skull-brain interface; *Pressure ratio* is determined by dividing the maximum pressure within the brain by the maximum pressure in the head (including scalp, skull and brain).

This table is also added to the Supplementary Information as Supplemental Table S1 in this revision.

7. the authors state that the skulls were acquired from Skulls Unlimited. Did the donor give full consent? Was it approved by the IRB of Carnegie Mellon University? Boston University?

Response: Thank you for your concern. We contacted the vendor and they have provided us with a HIPAA redacted form in which an authorized family member of the deceased donor consented to donate their whole body for education, science, and/or research purposes.

Our IRB (approved by Advarra, as noted in the manuscript) approved our *ex-vivo* experiment procedure.

Reviewer #2 (Remarks to the Author):

The authors performed an extensive revision and response to the raised points. This allowed to clarify several aspects, but could not address all of the important aspects. The additional analyses were very helpful, but opened or raised novel critical points.

Response: We are grateful that the reviewer finds our additional analyses helpful, and have addressed all remaining or new points raised by the reviewer, detailed below.

The authors changed now completely the introduction, overall I think it fits clearly better to the purpose of the study than the previous version of it. In my view it is a bit lengthy and repetitive in certain points. The authors state here that V5 is the main/primary brain area processing motion ‘The primary brain area associated with visual motion processing is V5...’. I do not think that this is correct, there is significant literature showing that V1 is already involved in motion detection/processing in interaction with V5, thus the analyses of the interaction of both would be also quite important.

Response: We are glad that the reviewer finds the revised introduction more fitting. Thank you for your excellent point about V1 – V5 interactions. This paper is not intended to get into the weeds of the argument of primary brain areas, and we certainly did not mean to belittle or offend any research looking into V1’s role in the visual motion processing pipeline. We have adjusted the language, slightly to, “**A critical brain area involved in** visual motion processing [...],” for which the citations we provided in the manuscript should offer ample support.

Overall the statistics is still not very clear or convincing for me. In e.g., Fig1 of the response letter there are still individual trials in the ANOVA included, in my view it clearly violates core assumptions of independence of an ANOVA. Furthermore, why is a one-tailed test used? I do not think that there were such strong assumption for all the conditions that one-tailed test are sufficient. Furthermore, in Fig. 3 for every frequency band different trial numbers are used? In the view of the quite small effect sizes also a Bayesian approach would probably provide a more clear picture in regard of the effects.

Response: We are sorry for not making our statistics more clear in the previous revision. The hypothesis tests were not conducted on the raw data, but rather the Linear Mixed Effect models. These take repeated measures into account. We have added the following statement:

“We analyzed the bulk of our data with linear mixed effect models. These account for both fixed effects (i.e. experimental condition) and random effects (i.e. subject to subject differences) and allow for analysis involving repeated measures^{118,119}.”

In each test, outliers were removed via the IQR test. We have also noted this in the manuscript:

“Data for each statistical test underwent interquartile range outlier rejection in Python, were further imported into R and fit with their specified equations (Equations 1 – 3). The post outlier rejection sample sizes are all noted on their relevant figure panels. Multiple comparison corrections were conducted as noted under each of the relevant sections.”

We have added a brief justification of our one-tailed hypothesis test usage in our new *Statistical Analysis* methods section.

“Due to previous tFUS research demonstrating ultrasound neuromodulation to enhance visual motion perception⁵¹ and amplify event related potential amplitudes^{50,126}, we had *a priori* hypotheses that tFUS-GC would enhance the visual-motion based BCI performance, and that it would amplify the relevant neural signals. As a result, we used one-tailed z-tests in our hypothesis testing for a lower Type II error rate.”

We want to thank the reviewer for the excellent suggestion to use a Bayesian statistical approach! We have run a Bayes Factor analysis and have included it in our results, discussion, methods, and Figure 1c. In short, the Bayes Factor analysis provides us with robust evidence that experimental condition has a moderate to strong effect on the Euclidean error, which is consistent with what we see with Cohen’s *d* and *p*-value analysis. We have provided Figure 1c below with the corresponding caption. Please find more specifics for results, discussion, or methods in the revised manuscript.

Figure 1c: The effect of experimental condition was also quantified using Bayes Factors (BF). To provide a more robust view of the effect, the median \pm 95% confidence interval (over 1,000 iterations) BF was plotted over a range of fixed effect scaling factors, which provides a high degree of confidence in that experimental condition has a moderate ($BF > 3^{62}$) to strong ($BF > 10^{62}$) effect on the Euclidean error. For the recommended default value (scaling

factor = 0.5), the BF was calculated to have a median of 14.0 and a 95% confidence interval from 13.8 to 14.4, which constitutes a strong effect.

Subjects and conditions: the authors state that due to time restrictions they prioritized conditions, thus the present trial was not a randomized trial as stated in the MS, but the conditions were prioritized and in a non-randomized order, which might have significantly influenced the findings, as factors like drowsiness or surprise, novelty at the beginning of the task, learning might play a significant role.

Response: Thank you for pointing this out. The reviewer is correct that the order of experimentation might not be considered 100% randomized given that we did prioritize testing tFUS and Non-Modulated conditions when time was limited. However, we randomized order as best we could given the subject time constraints mentioned in our previous responses. Our linear mixed effect model accounts for potential drowsiness/novelty/learning as best we could by introducing the additional *order * scans* term in our linear mixed effect model.

We have included a discussion on this in our Methods section, copied below for your convenience:

“In some cases, the approved 3-hour experiment time did not allow for testing all four conditions and subjects were unable to be scheduled back. When it was apparent that time would become an issue, we prioritized testing tFUS-GC and Non-modulated conditions, given that this project was majorly to test whether tFUS neuromodulation could enhance BCI performance. The partial data for these sessions were included in analysis if they did not meet the other exclusion criteria.

Given that we did prioritize the testing of tFUS-GC and Non-modulated conditions over the other two controls, the experiments may not be considered truly 100% randomized. We account for this, as the best we can, in our statistical analysis (see the *Linear mixed effect models* subsection of *Statistical Analysis* for details).”

Linear mixed effect models

[...] We also account for the only predominantly randomized condition in these models, by starting with including a factor in them for the order in which the condition was tested (see Equation 1). This term was not a significant factor in the neural analysis models (based on a Type III ANOVA), so it was omitted.

I still do not fully agree with the authors’ selection of comparisons of conditions. I guess they had an a priori hypothesis which indicated an statistical analysis to test the primary hypothesis and I guess it included here all control conditions in the hypothesis and not only verum vs sham? Thus the argumentation in terms of the used contrasts is not so clear. This is especially important in term of the small effect size.

Response: We appreciate this excellent comment. We had more selective comparisons in our previous draft (tFUS-GP vs. specifically tFUS-GC, tFUS-GC vs. no active tFUS), but we were strongly encouraged and advised by the reviewer to switch all tests to include comparisons across all four conditions for every test. The range and robustness of our control conditions do result in a number of significant differences amongst themselves, but in the tests, there are still strong contrasts between tFUS-GC and the control conditions.

The authors provided nicely more details about the different conditions, especially about the tFUS-GP condition. In the view of the analyses of the activity provided in Fig. S1, one gets the impression that the beam of the tFUS-GP (2cm from hotspot) is actually still in the activated area.

Response: Thank you for the excellent question. For visualization purposes, Figure S1 was an inflated brain model, which shows a continuous surface of unfolded sulci and gyri. The unfolded model may very well have a hotspot that goes on for more than 1.41 cm, however, the actual human brain is folded. Figure 6d includes a diagram of the same brain (the FreeSurfer *FSAverage* model) in the folded state, with area V5 highlighted in yellow. In Figure 6d, the green X corresponds to 1.41 cm away, and is outside even the structural area of V5. We have included some subject-specific visualizations of the active site with an annotated tFUS-GP targeting site as Supplemental Figure S4, which shows that the tFUS-GP targeting site (marked with a red X) was steered beyond the subject-specific functional areas.

Supplemental Figure S4: Subject-specific targeting maps for tFUS-GP. The N200 timeframe was imaged for four subjects during the non-modulated BCI task. Area V5 is highlighted in purple and outlined in black. The reconstructed EEG source activities are colored from red to yellow-white. A 1.41-cm black dashed line (corresponding to the steering length of tFUS-GP) is started at the approximate geometric center of V5. The other end of the dashed line, marked with a red X, corresponds to the target location of tFUS-GP. In all cases, tFUS-GP is applied to the structural periphery of area V5, but beyond the main functional area.

The authors provided further analyses in other important frequency bands for such a task e.g., the alpha, beta and gamma band (Fig. 3). From these new analyses, the effects induced by tFUS appears to be rather unspecific in terms of frequency with differences between all different conditions and not only between the verum condition and the control. This creates quite relevant doubts in terms of the physiological meaning or specificity of the physiological meaning of the

tFUS impact on brain oscillatory activity in the visual system. Also here the n for every single condition and frequency band is different? What is the reason for this? Artefact rejection? Should it not be the same? Also here in the view of the multitude of posthoc testing and that the authors apparently did not have a frequency specific hypothesis in my view one-tailed testing is not an adequate approach. What results would two-tailed testing have provided?

Response: Thank you for the excellent comment. At the site of sonication, V5, there are indeed quite a few substantial differences across conditions. However, particularly, we are interested in the specific comparisons in tFUS-GC vs. the control regions, and in the theta-alpha frequency ranges, tFUS-GC N200's power is significantly amplified over the other three controls. Further, downstream the processing pathway in SP, the theta-alpha ranges show significant increases for tFUS-GC compared to all three conditions, and no other significant differences are observed. We believe these data are rigorous and convincing.

We have further clarified the reviewer's questions of sample sizes and one-tailed testing in our new Statistical Analysis methods subsection. In short, we performed outlier exclusion for each processed dataset before hypothesis testing, and prior tFUS research have provided us with *a priori* hypotheses with respect to the behavioral and neural responses to tFUS.

For one-tailed tests:

“Due to previous tFUS research demonstrating ultrasound neuromodulation to enhance visual motion perception⁵¹ and amplify event related potential amplitudes^{50,126}, we had *a priori* hypotheses that tFUS-GC would enhance the visual-motion based BCI performance, and that it would amplify the relevant neural signals. As a result, we used one-tailed z-tests in our hypothesis testing for a lower Type II error rate.”

For sample size, we clarify, in our new Statistical Methods section:

“Data for each statistical test underwent interquartile range outlier rejection in Python, were further imported into R and fit with their specified equations (Equations 1 – 3). The post outlier rejection sample sizes are all noted on their relevant figure panels. Multiple comparison corrections were conducted as noted under each of the relevant sections.”

The different pulsing might have influenced the results. Did the authors compare subjects with 1, 2, 4 times pulsing? This might have led to distraction, fatigue etc in an incoherent way?

Response: Thanks for the great suggestion. We have compared the average Euclidean error for each subject in the tFUS-GC condition, based on how many sonications they received (Figure III). Our data yields no statistical difference (Kruskal-Wallis; $p = 0.425$) regarding sonication numbers. We do see a negative trend, where error seems to visually decrease with sonication number, but, possibly due to our limited sample sizes for the smaller pulse number range, this difference is not significant.

Figure III. Average tFUS-GC Euclidean error as a function of number of sonications. There was no significant difference (Kruskal Wallis test, $p = 0.425$) found for error as a function of sonication number.

In this revision, we added some discussions in this topic, and it would be another interesting topic to conduct a future study looking specifically at effect of sonication number.

“As noted in the methods section, we applied between one and four 500 ms sonications to each subject per BCI epoch. We analyzed the differences in corresponding subject-average tFUS-GC Euclidean error as a function of number of sonications and found no significant difference (Kruskal-Wallis; $p=0.425$). This may be due to the skewed sample sizes (18 of the 25 subjects received four sonications per epoch), and future studies are warranted to investigate the exact effect and optimization of sonication number.”

The authors provided an informative overview of the activity in the visual cortex during the task (Fig. S1) and how the focus of the tFUS beam was. It shows that the beam is in the activation area, however, it suggests also that the tFUS-GP control condition might actually have also reached the relevantly activated areas. Furthermore, this is a group average with rather widespread activity there might be relevant differences in individual subjects, that might lead to heterogeneity. Thus it would be interesting to see individual overlaps between the sources and the tFUS beam of individual subjects.

Response: Thank you for the additional comment on tFUS-GP targeting. As we mentioned above, the Fig S1 is of an inflated brain model, so the total distance of the source is somewhat overstated. We have included some subject-specific brain maps with annotated tFUS-GP targeting relative to the non-modulated functional zone in our supplementary information:

Supplemental Figure S4: Subject-specific targeting maps for tFUS-GP. The N200 timeframe was imaged for four subjects during the non-modulated BCI task. Area V5 is highlighted in purple and outlined in black. The reconstructed EEG source activities are colored from red to yellow-white. A 1.41-cm black dashed line (corresponding to the steering length of tFUS-GP) is started at the approximate geometric center of V5. The other end of the dashed line, marked with a red *X*, corresponds to the target location of tFUS-GP. In all cases, tFUS-GP is applied to the structural periphery of area V5, but beyond the main functional area.

Further, while it is possible that tFUS-GP may have activated some small fraction of the active site in some participants (Supplemental Figure S4), on average, it did not lead to significantly different BCI performance metrics, and there are still ample differences in power, connectivity, and cross-frequency interactions (please see below) between it and tFUS-GC, which also supports the tFUS-GP was targeted away from the active site.

I think it would be of large interest in the present data to look for cross-frequency interactions, especially in the view of the results of the new analyses that do show changes in all frequency bands. It would help to better understand the specificity of the tFUS effects here.

Response: We agree with the reviewer that further exploration of cross-frequency analysis may make for an extremely interesting follow-up tFUS study. However, typically, visual-system cross-frequency analysis investigate the coupling to high gamma (> 40 Hz) [1, 2]. This is done with invasive arrays (ECoG in humans) that are able to get “clean” high frequency recordings. However, standard scalp EEG practices involve band-pass filtering from 1 – 40 Hz, so acquiring information about 40+ Hz frequency coupling is challenging. Future research could investigate the coupling to high gamma in NHP or consenting humans with invasive electrode implantations. Nevertheless, as the reviewer suggested, we have performed the analysis suitable to EEG capabilities. The results do provide some limited support that tFUS-GC increases attention, but we do not think they are compelling enough to include in our updated manuscript or supplemental information.

We have conducted cross-frequency analysis for theta-alpha (4 – 12 Hz) to frequencies in the 4 – 40 Hz range coupling at V5, the superior parietal lobe, and the inferior temporal gyrus (Figure IV). Chacko et al. [2] did provide analysis of theta-alpha to beta-low-gamma (TABL; 12 – 40 Hz) coupling and found that increases in this metric was negatively correlated with reaction times (i.e., increased coupling increased reaction speed). This would suggest that increased TABL activity is associated with increased attention for behavioral tasks. We have visually identified clusters of increased TABL coupling compared to the non-modulated condition based on tFUS targeting location. For tFUS-GC, there are increased TABL PAC clusters in SP (Figure IVa) and V5 (Figure IVb), which were not present in the tFUS-GP or decoupled-sham conditions. In the tFUS-GP condition, which was sonicated to the periphery of V5 near the border of IT, there was increased TABL PAC in area IT (Figure IVc). This increased cluster was not present in the tFUS-GC or decoupled-sham conditions.

Our findings of increased TABL clusters in the tFUS-GC condition at V5 and SP are consistent with Chacko et al’s findings of increased TABL leading to increased behavioral-task based attention. However, there are also ample negative and positive PACs that are visually concordant across conditions, which may raise additional questions about tFUS specificity results. Some of this may be due to the complexity of the BCI speller; the task did not just involve visual motion perception, but keyboard colors and letters, too. **Future work** investigating tFUS’ specific effects on cross-frequency coupling is warranted, and we recommend that such work use one visual stimulus paradigm at a time, such as just motion, or just color, or just letters, in order to more confidently parse out the effects.

Figure IV. Averaged phase-amplitude coupling (PAC) difference from the non-modulated response. The subject-average PAC value was computed for each of the experimental conditions, and averaged over 100 random states. The differences between tFUS-GC, tFUS-GP, and decoupled-sham from the non-modulated condition was computed (Δ PAC) and filtered with a Gaussian kernel. **a)** At the superior parietal lobe, there is a noticeable increase of theta-beta coupling in the tFUS-GC condition compared to the non-modulated. This increase is not present for the tFUS-GP or decoupled-sham conditions. **b)** At V5, there is increased theta-alpha coupling to 35+ Hz gamma in tFUS-GC, which is less expressed in the two tFUS control conditions. **c)** At the inferior temporal gyrus, there are more pronounced increases of coupling in the theta-alpha range to gamma activity in tFUS-GP condition compared to the tFUS-GC or decoupled-sham conditions.

[1] Szczepanski, S. M. *et al.* Dynamic Changes in Phase-Amplitude Coupling Facilitate Spatial Attention Control in Fronto-Parietal Cortex. *PLoS Biol.* **12**, e1001936 (2014).

[2] Chacko, R. V. *et al.* Distinct phase-amplitude couplings distinguish cognitive processes in human attention. *NeuroImage* **175**, 111–121 (2018).

The discussion does not provide a clear explanations and concept to explain the changes in basically all conditions and all frequency bands. It is probably also very difficult to define a general

conceptual framework of the tFUS effect as the behavioral effects are quite small and the electrophysiological results based on the new analyses seem to be quite unspecific.

Response: With all of the added frequency band analysis, we apologize if the main finding of the paper has gotten muddled. This paper is, first and foremost, the first paper to investigate whether tFUS to the center of V5 can enhance BCI outcomes. Our results support this, both in terms of traditional p -values and in terms of our newly added Cohen's d quantification and Bayes Factor.

We **also** provide a general underlying conceptual framework for why this might be, by investigating the induced neurological differences between tFUS-GC and the three control conditions. While the beta and gamma bands are somewhat unspecific, throughout the theta and alpha bands, these results are quite specific (tFUS-GC is significantly amplified over all three of the control conditions). This is why we focus our discussions on those ranges, weaving together our results in the neuroscience canon.

The three controls having different effects on the underlying neurological response **is not unsurprising**. The decoupled-sham has audio, which can be distracting and effect the magnitude of ERPs, as we mention in the discussion, and tFUS-GP is applying ultrasound to another part of the brain. We are just applying tFUS to a visual motion stimulus; the BCI has colors, letters, shapes, etc. Applying tFUS-GC may inadvertently change the brain's feature-based attention to one or more of these items. This is beyond the purview of our current project. This project, and the contents of this paper, are investigating how tFUS to the geometric center of V5 can effect BCI outcomes, for which there are significant differences between tFUS-GC and all three control conditions, and there is significantly amplified theta and alpha N200 power between tFUS-GC and all three of the control conditions at V5 and SP.

We have included more emphasis on the behavioral results in our concluding paragraph, to reiterate that the main finding of the paper is a behavioral one.

“In conclusion, our results are the first to demonstrate that tFUS neuromodulation to human V5 significantly reduces errors in mVEP-based BCI spelling. This is quantified with traditional p -value analysis, Cohen's d effect size, and Bayesian Factor Analysis, all of which concur that the effect of tFUS neuromodulation to the V5 is medium to strong. We also investigate the underlying neurological changes induced by tFUS and sham conditions to provide a thorough framework for the underlying mechanism. Concurrent neural electrophysiological recordings indicate that in the mVEP-based BCI, tFUS neuromodulation causes a significant amplification of theta and alpha powers in the immediate V5-targeted area, as well as downstream in the dorsal vision processing pathway. Connectivity analysis, along with previous research on alpha-theta power, provides critical evidence that the increased theta power due to concurrent tFUS neuromodulation may lead to heightened feature-based attention to visual motion stimuli.”

REVIEWER COMMENTS

Reviewer #3 (Remarks to the Author):

In the manuscript “Transcranial Focused Ultrasound to V5 Enhances Human Visual Motion Brain-Computer Interface by Modulating Feature-Based Attention” the authors investigate whether noninvasive neuromodulation can improve BCIs. They find that tFUS improved BCI outcomes in a speller task and relate this improvement to theta and alpha activity in V5 and downstream visual pathway regions. In the previous reviews concerns regarding the data analysis were brought up and addressed by the authors, so I will focus on that aspect of the manuscript.

Main points

1.

Figure 1 and related analyses (but the same points apply to other analyses and figures as well)

In (a) I presume each dot shows shows the Euclidean error in a single trial, and this is pooled over subjects. The vertical distributions seem to be some smoothed fit to those data points. Please clarify in the caption what is shown here exactly. I would plot the data points on top of the quartile boxes so that they are visible despite the overlap.

Outliers are removed using the 1.5 quartile ranges. This method is suitable for roughly Gaussian distributed data, but here there are very non-Gaussian distributions with a long tail, which makes the outlier removal problematic in my opinion.

The way the results of the ANOVA tests are visualised in the figure strongly suggests that they were done directly on these distributions, which is confusing or potentially misleading. The authors explain in the text that the tests are actually performed on the coefficients of the mixed models, but do not provide much detail on this besides stating this.

The corresponding description related to the analysis using the mixed models is not very clear in the Methods. In equation 1 the notation is close to, I think, how the code looks in R, but for the reader additional information is required. So the Euclidean Error is predicted using the predictors Condition, Scans, Order:Scans, and the individual subject. How are each of those predictors represented (e.g. with dummy variables for each category level)? The result of the fit, I guess, is a coefficient (beta), so to test the influence of Condition, you then obtain a different coefficient for each condition (i.e. stimulation, sham etc.)? Then I guess you ran the ANOVAs etc. on those model coefficients (see line 638)? Please clarify and elaborate on this process. Maybe I am missing something here, but wouldn't you then just

receive one beta coefficient for a given condition? If so how do you compare them using the ANOVA? I think these aspects need all to be described properly in the Methods and Results as the main result is based on these analyses.

The same issues (outlier removal, visualisation, details on model fits and statistics) apply to Figures 3 and 4 and the corresponding descriptions in the Methods and Results.

2.

In Figure 1b Cohen's d is presented as the effect size of the BCI outcomes. I would read that as that as if the stimulation improves the EE by ~0.5 to 0.7 standard deviations. However, this does not seem to be the case because the Cohen's d seems to be calculated based on the model coefficient distributions. Therefore, it does capture the rather abstract change in the fitted model parameters but not the directly the actual improvement in the EE, which could be potentially misleading for the reader.

3.

Line 114: For the analyses in Fig. 1 a Bonferroni correction for multiple testing is applied. For other analyses a False Discovery adjustment is used (e.g. Fig. 3 and 4). The Bonferroni correction is considered to be overly conservative, i.e. at risk of leading to many false negatives. Why were different types of adjustment used for different analyses?

4.

Figure 2c (right panel): The figure legend obscures the baseline period. There seem to be differences between conditions of similar magnitude as the emphasised differences at 160-230ms.

Minor points

Line 130 (same elsewhere): the superscripted references should be e.g. before the brackets, so they do not look like exponents.

Figure 3 (and main text description): Why is it Arbitrary Units (a.u.) on the y-axis? As far as I understood the processing, the values should be z-scores?

Equation 8 (and similar in others): I am not sure about the use of \approx here (the approximately equal symbol). For the Bayesian estimates the denominator can be ignored here because e.g. $P(B)$ is effectively

a constant. I think this just means that for the inference we do not need to do this normalisation using $P(B)$, as it would just scale all values so that the result is still a probability between 0 and 1. Without the scaling this is no longer given, so the actual values are different (thus I am not sure about the ' \approx '), but it does not matter for the inference. I did not understand the sentence in line 688-689; how can a Euclidean Error be taken as a probability?

Response to Reviewer Comments

We are grateful to the reviewer for her/his helpful and constructive comments. We have conducted additional work to address all reviewer comments and revised the manuscript accordingly. We have addressed all the comments point-by-point in blue in this document. Revised texts are printed in RED in the revision. Figures 1, 3 and 4 are updated in the main manuscript, Supplemental Table S2 is added to Supplementary Information. Based on these further clarifications and validations on statistical analyses, we believe the revised manuscript has been further improved and represents a major contribution to the field for the very first transcranial focused ultrasound study enhancing human visual motion brain-computer interface by modulating feature-based attention.

Reviewer #3 (Remarks to the Author):

In the manuscript “Transcranial Focused Ultrasound to V5 Enhances Human Visual Motion Brain-Computer Interface by Modulating Feature-Based Attention” the authors investigate whether noninvasive neuromodulation can improve BCIs. They find that tFUS improved BCI outcomes in a speller task and relate this improvement to theta and alpha activity in V5 and downstream visual pathway regions. In the previous reviews concerns regarding the data analysis were brought up and addressed by the authors, so I will focus on that aspect of the manuscript.

Main points

1.

Figure 1 and related analyses (but the same points apply to other analyses and figures as well)

In (a) I presume each dot shows shows the Euclidean error in a single trial, and this is pooled over subjects. The vertical distributions seem to be some smoothed fit to those data points. Please clarify in the caption what is shown here exactly. I would plot the data points on top of the quartile boxes so that they are visible despite the overlap.

Response: Thank you for the excellent suggestion. We have increased the visibility of the raw data points by shifting them to the right of the box plots and updated the caption for improved comprehension. Please see the updated versions below:

Figure 1. tFUS to V5 significantly improves mVEP BCI speller outcomes. Data were fit to a linear mixed effect model to account for repeated measures across subjects and additional fixed effects from learning and fatigue (Equation 1). a) A raincloud plot of the BCI Euclidean error. The left portion of each subplot is a violin plot highlighting the distribution of the data. A box plot is at the center, marking the data median (center line), 1st and 3rd quartiles (whiskers = 1.5 * quartile ranges) and means (black triangle). The raw datapoints are presented to the right. Data are analyzed with the results of the linear mixed effect models. An ANOVA and a one-tailed z-test with Bonferroni correction were performed on the model's fit for each condition. [...]

Outliers are removed using the 1.5 quartile ranges. This method is suitable for roughly Gaussian distributed data, but here there are very non-Gaussian distributions with a long tail, which makes the outlier removal problematic in my opinion.

Response: Thank you for this comment. We have expanded the justification for IQR outlier rejection in our methods section. In addition, we re-ran the tests for all of our major findings using double MAD outlier rejection, and found the results to be consistent.

“Outlier rejection

Outliers were identified and discarded using the interquartile range method (Equation 5). The interquartile range (IQR) is the difference between 75th percentile (Q3) and 25th percentile (Q1) of the data, and data are considered outliers if they fall 1.5 * IQR below Q1 or 1.5 * IQR above Q3. IQR rejection has been found to control for outliers even in the case of skewed data¹¹⁶, and is widely used in biomedical sciences with non-normally and asymmetric distributed data^{117,118}.

To verify our results, we re-ran all of the tests for our major results using the double median absolute deviation (double MAD)¹¹⁹ outlier rejection strategy. Data were considered outliers if they were beyond 3.5 double MAD from the median. All the statistical outcomes were validated as consistent (Supplemental Table S2).”

The way the results of the ANOVA tests are visualised in the figure strongly suggests

that they were done directly on these distributions, which is confusing or potentially misleading. The authors explain in the text that the tests are actually performed on the coefficients of the mixed models, but do not provide much detail on this besides stating this.

Response: Thank you for your excellent point. We have further clarified this point in both the caption and the figure (through recoloring of the significance labels and adding the statement “Hypothesis testing performed on linear mixed effect models” to the image itself). Please see our changes below:

Figure 1. tFUS to V5 significantly improves mVEP BCI speller outcomes. Data were fit to a linear mixed effect model to account for repeated measures across subjects and additional fixed effects from learning and fatigue (Equation 1). a) A raincloud plot of the BCI Euclidean error. The left portion of each subplot is a violin plot highlighting the distribution of the data. A box plot is at the center, marking the data median (center line), 1st and 3rd quartiles (whiskers = 1.5 * quartile ranges) and means (black triangle). The raw datapoints are presented to the right. Data are analyzed with the results of the linear mixed effect models. An ANOVA and a one-tailed z-test with Bonferroni correction were performed on the model's fit for each condition. [...]

The corresponding description related to the analysis using the mixed models is not very clear in the Methods. In equation 1 the notation is close to, I think, how the code looks in R, but for the reader additional information is required. So the Euclidean Error is predicted using the predictors Condition, Scans, Order:Scans, and the individual subject. How are each of those predictors represented (e.g. with dummy variables for each category level)? The result of the fit, I guess, is a coefficient (beta), so to test the influence of Condition, you then obtain a different coefficient for each condition (i.e. stimulation, sham etc.)? Then I guess you ran the ANOVAs etc. on those model coefficients (see line 638)? Please clarify and elaborate on this process. Maybe I am missing something here, but wouldn't you then just receive one beta coefficient for a

given condition? If so how do you compare them using the ANOVA? I think these aspects need all to be described properly in the Methods and Results as the main result is based on these analyses.

Response: Thank you for the comment. The notation in the current equations is indeed based on R's equation formatting. We have clarified the mathematic formula in our methods section:

Mathematically, this is equivalent to

$$EE_{i,k} = \beta_0 + \beta_{1,j} * Condition_j + \beta_2 * Scans_i + \beta_3 * Order_j + \beta_4 * (Scans_i * Order_j) + b_{k,i} + \epsilon_{i,k} \quad (2)$$

Where i corresponds to the i th trial of subject k , and j denotes the experimental condition of trial i . β_1 is a (1 x J) vector of the weights for each condition, and $Condition$ is a (J x 1) one-hot vector. $\beta_0, \beta_2, \beta_3$ and β_4 are scalar quantities. $Scans_i$ and $Order_j$ are discrete integer values corresponding to the number of repeat scans of trial i and the order in which condition j was tested, respectively. $b_{k,i}$ is the random-effect of the subject k for trial i , defined as a Gaussian distribution with a subject-specific mean and variance. $\epsilon_{i,k}$ is the error term.

We have also clarified how the intra-condition comparisons are made based on R's functions:

To conduct z-tests comparing each of the J experimental conditions (in our case, $J = 4$), linear mixed effect models produce estimates of the true mean β (a $J \times 1$ vector) and covariance Σ (a $J \times J$ matrix) of the experimental conditions as a multivariate mean $\hat{\beta}$ (a $J \times 1$ vector; Equation 6) with some covariance $\hat{\Sigma}$ (a $J \times J$ matrix). These estimations are performed with R's lme4¹²³ function *lmer*. These values are then used by R's multcomp¹²⁴ function *glht* to compute z-statistics using Tukey's all pair-wise comparisons. For an in-depth description, please refer to Hothorn et al¹²⁸. To get the pooled standard error of any two conditions l and m (SE_{lm}), the square root of the variance of condition l ($\hat{\Sigma}_{ll}$) plus the variance of condition m ($\hat{\Sigma}_{mm}$) minus two times the covariance of conditions lm ($\hat{\Sigma}_{lm}$) are calculated (Equation 7). The estimated means for condition l ($\hat{\beta}_l$) and m ($\hat{\beta}_m$), along with the pooled standard error, are used to compute a paired z-test statistic (Equation 8). Multiple comparison corrections was conducted as noted under each of the relevant sections.

$$\hat{\beta} \sim N(\beta, \Sigma)$$

$$SE_{lm} = \sqrt{\hat{\Sigma}_{ll} + \hat{\Sigma}_{mm} - 2 * \hat{\Sigma}_{lm}}$$

$$z_{lm} = \frac{\hat{\beta}_l - \hat{\beta}_m}{SE_{lm}}$$

The same issues (outlier removal, visualisation, details on model fits and statistics) apply to Figures 3 and 4 and the corresponding descriptions in the Methods and Results.

Response: Thank you again for your comment for clarification. We have updated our figures and captions for Figures 3 and 4 in the same way as we updated Figure 1 above, with a recolored statistical analysis (navy blue) and a statement that statistical testing was performed by LMMs in the same color.

2.

In Figure 1b Cohen's d is presented as the effect size of the BCI outcomes. I would read that as that as if the stimulation improves the EE by ~ 0.5 to 0.7 standard deviations. However, this does not seem to be the case because the Cohen's d seems to be calculated based on the model coefficient distributions. Therefore, it does capture the rather abstract change in the fitted model parameters but not the directly the actual improvement in the EE, which could be potentially misleading for the reader.

Response: Thank you for your excellent comment. We have clarified a few points in our updated manuscript, including why the captured change is not so abstract. The values returned by R's *glht* function are the estimated differences in means between experimental conditions, and the pooled error between them. The estimates are based on the linear effect model, but, the models provide a more robust estimate of the effect and errors for each condition than pooling all the raw data together, as we do have repeated measures and not all the subjects were able to test every condition. These values, differences in mean of condition effect and pooled error (converted to standard deviation) of the data, are exactly what Cohen's d uses in the calculation. The following has been added to our Methods section:

“To quantify the effect size, we calculated Cohen's d (Equation 9) based on the fit linear mixed effect models. R's *general linear hypothesis testing (glht)* function directly returns the estimated difference in means ($\mu_1 - \mu_2$) and the pooled standard error of the means (SE) for each condition-condition comparison. It is worth noting that these values are not calculated directly from the raw data of the experiment, which, as mentioned above, are also impacted by repeated measures and additional factors like the scan number and order of testing, but instead from the linear mixed effect model's estimates of the true effect on the mean and standard error for each condition. To convert the SE to pooled standard deviation (σ_{pooled}), we multiplied the SE by the square root of the subject number.”

3.

Line 114: For the analyses in Fig. 1 a Bonferroni correction for multiple testing is applied. For other analyses a False Discovery adjustment is used (e.g. Fig. 3 and 4). The Bonferroni correction is considered to be overly conservative, i.e. at risk of leading to many false negatives. Why were different types of adjustment used for different

analyses?

Response: Thank you for your excellent comment. We chose to use Bonferroni correction for the BCI error because the BCI outcome is the main point of our paper. As such, we wanted to take extra precautions to avoid potential false positives in the behavior analysis. For the neural data analyses, we adopted a less conservative approach because we wanted to provide a possible mechanistic understanding of how tFUS may have improved BCI outcomes. We have clarified this point in our manuscript's discussion:

“To address the multiple comparisons problems, we used Bonferroni and false-discovery rate (FDR) multiple comparison corrections. We used Bonferroni correction, which is more conservative, to rigorously investigate the BCI behavior error outcomes, as effect of tFUS on BCI performance was the primary focus of this investigation and we wanted to be extra conservative in avoiding potential false positives (type I errors). For the neural data analysis, we used FDR, which is a less conservative method that can better balance the risk of possible type I and type II errors.”

4.

Figure 2c (right panel): The figure legend obscures the baseline period. There seem to be differences between conditions of similar magnitude as the emphasised differences at 160-230ms.

Response: Thank you for the comment. We have updated the figure to move the legend outside of the plotting box. The permutation cluster test did not result in any other significant time windows for those occipital electrodes. Please see the updated figure below:

Figure 2. Significant differences are found between conditions in the EEG sensor domain. a) Averaged left posterior electrode responses for 1 to 40 Hz (graph column 1), theta frequencies (graph column 2), alpha frequencies (graph column 3), beta frequencies (graph column 4), and gamma frequencies (graph column 5) for tFUS-GC (top), decoupled-sham (second row), non-modulated condition (third row), and tFUS-GP (bottom). Yellow and purple highlight the approximate N200 and P300 waveform responses within the 100 to 250 ms and 250 to 400 ms windows, respectively. b) Topographic maps of the trial's averaged activity for 0 to 100 ms (left), 100 to 250 ms (middle) and 250 to 400 ms (right) post stimulus filtered between 1 to 40 Hz for tFUS-GC (top), decoupled-sham ("DS"; second row), non-modulated ("NM"; third row), and tFUS-GP (bottom) conditions. The topo colormap is scaled to the relative max/min response for each. c) A significant spatiotemporal cluster ($p < 0.05$) between the 1 to 40 Hz filtered conditions using MNE Python's nonparametric spatiotemporal cluster test with 1000 permutations (test: repeated measures ANOVA) when comparing all four conditions. (left) The F-statistics of a significant spatial cluster denoted by white circles over the electrodes. (right) The mean activity across channels and trials of the four trial conditions (± 1 standard deviation). Gray shaded regions indicate a significant ($p < 0.05$) temporal cluster corresponding to the spatial cluster.

Minor points

Line 130 (same elsewhere): the superscripted references should be e.g. before the brackets, so they do not look like exponents.

Response: Thank you for the comment. We have reformatted the references outside the brackets to avoid confusion.

Figure 3 (and main text description): Why is it Arbitrary Units (a.u.) on the y-axis? As far as I understood the processing, the values should be z-scores?

Response: Thank you for this comment. We have removed the arbitrary units from the y-axis. We have also replaced "a.u." in the results section with "z-scores".

Equation 8 (and similar in others): I am not sure about the use of \approx here (the approximately equal symbol). For the Bayesian estimates the denominator can be ignored here because e.g. $P(B)$ is effectively a constant. I think this just means that for the inference we do not need to do this normalisation using $P(B)$, as it would just scale all values so that the result is still a probability between 0 and 1. Without the scaling this is no longer given, so the actual values are different (thus I am not sure about the ' \approx '), but it does not matter for the inference. I did not understand the sentence in line 688-689; how can a Euclidean Error be taken as a probability?

Response: Thank you for your excellent point about the *approximately equal* symbol. We have replaced it with the proportional to (\propto) symbol, which would be more appropriate.

For the Euclidean error as a probability, we have clarified this point in the methods as well:

“The probability of the typed letters given the corrected word was defined as 1 minus the Euclidean error between the two. This provides a similarity score between 1 (perfect match) and 0 (maximum possible error) between the typed letters and corrected word, which is deemed as a proxy of the conditional probability.”